# A dual role of Cohesin in DNA DSB repair

Michael Fedkenheuer [1] ✉, Yafang Shang[2], Seolkyoung Jung [1], Kevin Fedkenheuer[3], Solji Park[1], Davide Mazza[4], Robin Sebastian [5], Hiroyuki Nagashima[1], Dali Zong[6], Hua Tan [3], Sushil Kumar Jaiswal [3], Haiqing Fu [5], Anthony Cruz[7], Supriya V. Vartak[1], Jan Wisniewski[8], Vittorio Sartorelli [9], John J. O'Shea[1], Laura Elnitski [3], Andre Nussenzweig[6], Mirit I. Aladjem [5], Fei-Long Meng [2] & Rafael Casellas [10]

Cells undergo tens of thousands of DNA-damaging events each day. Defects in repairing double-stranded breaks (DSBs) can lead to genomic instability, contributing to cancer, genetic disorders, immunological diseases, and developmental defects. Cohesin, a multi-subunit protein complex, plays a crucial role in both chromosome organization and DNA repair by creating architectural loops through chromatin extrusion. However, the mechanisms by which cohesin regulates these distinct processes are not fully understood. In this study, we identify two separate roles for cohesin in DNA repair within mammalian cells. First, cohesin serves as an intrinsic architectural factor that normally prevents interactions between damaged chromatin. Second, cohesin has an architecture-independent role triggered by ATM phosphorylation of SMC1, which enhances the efficiency of repair. Our findings suggest that these two functions work together to reduce the occurrence of translocations and deletions associated with non-homologous end joining, thereby maintaining genomic stability.

Mammalian cells suffer an estimated 70,000 DNA-damaging events per day, resulting in single-strand DNA breaks (SSBs) and, less frequently, DSBs[1]. DSBs and defects in DSB repair underlie genomic instabilities that cause cancer and genetic disorders[1–5]. Cohesin, a multi-subunit protein complex, plays a significant role in the DNA damage response (DDR)[6]. Initially described as a driver of sister chromatid cohesion[7,8], cohesin also regulates the topology of chromosomes by extruding chromatin through its ringlike structure to form DNA loops[9]. These loops influence nearly all cellular processes by modulating the interaction frequency between DNA elements that may be separated by hundreds of kilobases. It has been proposed that loop extrusion enhances DNA repair by organizing local chromatin domains[10]. However, the underlying mechanism has not been clearly defined.

The cohesin ring consists of SMC1, SMC3, RAD21 and SA subunits[11–14]. In response to DNA damage, SMC1 is phosphorylated at serines 957 and 966, a modification essential for checkpoint signaling and repair[15,16]. This phosphorylation is catalyzed by the ataxia

[1]Molecular Immunology and Inflammation Branch, National Institute of Arthritis and Musculoskeletal and Skin Diseases, National Institutes of Health, Bethesda, MD, USA. [2]Key Laboratory of RNA Innovation, Science and Engineering, Shanghai Institute of Biochemistry and Cell Biology, Center for Excellence in Molecular Cell Science, Chinese Academy of Sciences, University of Chinese Academy of Sciences, Shanghai 200031, China. [3]Translational and Functional Analysis Branch, National Human Genome Research Institute, National Institutes of Health, Bethesda, MD, USA. [4]Experimental Imaging Center, Istituto di Ricovero e Cura a Carattere Scientifico (IRCCS) San Raffaele Scientific Institute, Milano, Italy. [5]Developmental Therapeutics Branch, Center for Cancer Research, National Cancer Institute, NIH, Bethesda, MD 20892-4255, USA. [6]Laboratory of Genome Integrity, National Cancer Institute NIH, Bethesda, MD, USA. [7]Translational Genetics and Genomics Section, National Institute of Arthritis and Musculoskeletal and Skin Diseases, National Institutes of Health, Bethesda, MD, USA. [8]EIB Microscopy and Digital Imaging Facility, National Cancer Institute NIH, Bethesda, MD, USA. [9]Laboratory of Muscle Stem Cells and Gene Regulation, National Institute of Arthritis and Musculoskeletal and Skin Diseases, National Institutes of Health, Bethesda, MD, USA. [10]Department of Hematopoietic Biology & Malignancy, Division of Cancer Medicine, The University of Texas MD Anderson Cancer Center, Houston, TX, USA. ✉e-mail: michael.fedkenheuer@nih.gov

telangiectasis mutated (ATM) kinase in response to DSBs[15] and by the ATR kinase in response to SSBs[15]. In addition to SMC1, ATM also phosphorylates cohesin's SMC3 and PDS5 proteins, leading to repression of transcription and DNA replication near DSBs[17]. This suggests that the cohesin complex may function as a general DNA repair factor. Consistent with this idea, patients with mutations in SMC1, SMC3, and the cohesin loader NIPBL exhibit increased sensitivity to DNA damage[18–21]. Despite these observations, a direct link between cohesin phosphorylation and DNA repair has yet to be established.

DNA damage promotes the recruitment of repair factors that form distinct foci around DNA breaks[22,23]. Foci represent highly dynamic microenvironments where repair factors interact with each other and with damaged chromatin, dispersing readily once repair is complete[23–26]. A key initiating event of repair foci is the phosphorylation of histone H2AX, which notably occurs within the confines of topological associated domains (TADs) in a contact-dependent manner[27]. This observation suggests a link between cohesin-mediated loop extrusion and γH2AX. However, the precise role of cohesin in the phosphorylation of H2AX remains unclear due to conflicting findings. Some early studies reported increased γH2AX upon cohesin depletion[28], while later research indicated that cohesin is required for H2AX phosphorylation[10]. Hi-C analysis revealed that TAD boundaries are strengthened in the presence of DNA breaks[10], and that TAD insulation increases across the genome of irradiated cells[29]. Conversely, more recent studies have argued that damaged chromatin aggregates in the nuclei of mammalian cells in response to ATM activity but does so independently of cohesin[30]. As a result, a consistent mechanistic understanding of cohesin's role in DNA repair remains elusive.

DNA DSBs are primarily repaired by non-homologous end joining (NHEJ). In this pathways, the MRN complex (comprising NBS1, Rad50, and MRE11) first senses the presence of breaks and activates ATM[31]. 53BP1 is then recruited and prevents DNA end resectioning[32,33]. Processed DSBs are then ligated by DNA Ligase 4[34]. NHEJ is essential not only for repairing random breaks but also for joining targeted lesions intermediate to antibody and T cell receptor production. Specifically, during VDJ recombination, RAG-induced DSBs are processed by the NHEJ machinery[35] and require cohesin[36]. Defects in NHEJ leads to aberrant repair and the formation of chromosomal translations. However, an understanding of the precise mechanisms underlying this activity is also elusive due to contradicting findings. On the one hand cohesin loss was shown to increase the likelihood of deletions and translocations[37] and replicative DNA stress[38]. On the other hand, recent work argued that loss of cohesin has no impact in the formation of translocations[30].

The discrepancies in the literature arise, in part, from the challenge of deconvoluting the role of cohesin in DNA repair from its function as an architectural protein. To address this, we investigated DNA damage and repair in cells either depleted of cohesin using a degron system or in cells with a phosphorylation-deficient SMC1A subunit. Our findings support a model in which cohesin fulfills two distinct, independent roles in repair. As an architectural protein, cohesin normally limits interactions between neighboring TADs and chromosomes, thereby reducing the incidence of cis-deletions and translocations during DNA damage. In addition, DSBs trigger an ATM kinase-mediated phosphorylation of cohesin, leading to a marked accumulation of cohesin around damaged chromatin. Surprisingly, this phosphorylation has little or no effect on local chromatin architecture. Instead, we demonstrate that this extrusion-independent function enhances repair speed, minimizing aberrant joining of distant breaks. We conclude that the two functions of cohesin work in tandem to ensure the error-free repair of DNA lesions in mammalian cells.

## Results

To dissect the role of cohesin in the repair of DNA breaks we expressed the restriction enzyme AsISI fused to the Estrogen Receptor (ER)[39], in HCT116 cells, allowing us to induce continuous site-specific DSBs upon addition of 4-hydroxytamoxifen (4OHT, Fig. 1a). Using MRE11 ChIP-Seq[40], we monitored both the location and extent of DNA damage upon AsISI translocation to the nucleus (Supplementary Fig. 1a). On average, we detected 80-85 AsISI DSBs per cell, six hours post 4OHT treatment (Supplementary Fig. 1a, b, Supplementary Data 1). Next, we examined strong DSB sites from a magnified viewpoint (30 kb resolution) and found no significant cohesin peaks directly at breaks unless they coincided with pre-existing CTCF or NIPBL peaks (Supplementary Fig. 1c). In some instances, we did observe diffuse cohesin signals above background levels around the break sites (~5-10 kb); however, this was most frequently associated with pre-existing NIPBL peaks near DSBs (Supplementary Fig. 1d, Supplementary Data 2).

Consistent with previous findings[41], there was an increase in cohesin signal around DNA DSBs up to ~1 Mb, as measured by RAD21 ChIP-Seq ($P < 0.01$ for all categories v Ctrl Fig. 1b). Recruitment around the damaged site measured using intrachromosomal distance from break categories: ≤10 kb, ≤250 kb, ≤1 Mb (fold change (FC) = 1.43, 1.34, 1.20 respectively), was diminished when ATM was inhibited (FC = 1.30 1.11, 1.02; $P$ = 0.03, 2E-12, <1E-16 WT vs. ATMi; Fig. 1b). As shown by genome-wide viewpoints of damaged chromatin visualized by MRE11 and γH2AX ChIP-Seq, cohesin accumulation occurred primarily at CTCF anchor sites within damaged domains (Fig. 1c). Cohesin anchor site accumulation was observed at high levels throughout damaged chromatin versus undamaged chromatin domains in WT cells (FC = 1.43, $P$ = 0.003; Fig. 1d).

A limitation of using a continuous DNA damage system is that DNA breaks and repair occur simultaneously. To increase the sensitivity of the assay, we deleted the LIG4 gene to prevent repair in G1 and allow DNA breaks to saturate (Supplementary Fig. 1e). With this strategy we were able to visualize a robust accumulation of repair factors. In the LIG4$^{-/-}$ background, the cohesin signal within γH2AX domains was increased 1.73-fold ($P$ < 1E-16 inside vs. outside; Fig. 1c, d), indicating that, similar to other repair factors, DNA repair limits the extent of cohesin recruitment to damaged chromatin.

To determine whether SMC1A phosphorylation impacts cohesin accumulation on damaged chromatin, we engineered phosphorylation-deficient SMC1A HCT116 cells (SMC1A$^{Pdef}$) by replacing serines 957 and 966 with alanines (Supplementary Fig. 1f). Consistent with previous findings in mice[42], SMC1A$^{Pdef}$ cells were viable and appeared phenotypically normal. Notably, we observed that cohesin accumulation at anchor sites within γH2AX domains was mostly dependent on SMC1A phosphorylation (FC = 1.17, $P$ = 0.02 inside vs. outside; Fig. 1c, d).

We then monitored the dynamics of ectopically expressed RAD21-GFP molecules in damaged cells by confocal microscopy. Using a 355 nm UV laser, we targeted DNA damage to a 2.5μm² area of ER-AsISI HCT166 nucleus sensitized with the Hoechst 33342 dye (Fig. 1e). Following UV laser treatment, cohesin accumulated locally in a linear fashion, to levels of 138% after 1 h (0.76 RFU/min, Fig. 1f). Notably, only a fraction of this recruitment appeared to be dependent on ATM kinase activity, as cohesin signals were reduced by just 18% in ATMi-treated cells compared to the untreated damage control (Fig. 1f). Conversely, accumulation was nearly abolished in SMC1A phosphorylation deficient cells (Fig. 1f). These findings might be best explained by the fact that UV laser damage induces both DSBs and SSBs, and that SMC1A is phosphorylated by the ATR kinase in response to SSBs[15].

Previous studies showed that chromatin contact frequency increases around DSBs[10,30]. To quantify this effect, we conducted HiC analysis and plotted the fold change in intra-TAD contact frequencies (4OHT/untreated) for damaged domains and those that were or were not damaged by AsISI. Remarkably, despite the 1.4-1.6-fold increase in cohesin recruitment following AsISI lesion induction, the increase in intra-domain interactions within damaged TADs was minimal—only 1.02-fold compared to control TADs ($P$ = 9E-5 WT, Fig. 2a). This was in stark contrast to the 0.81-fold change in intra-TAD contacts observed

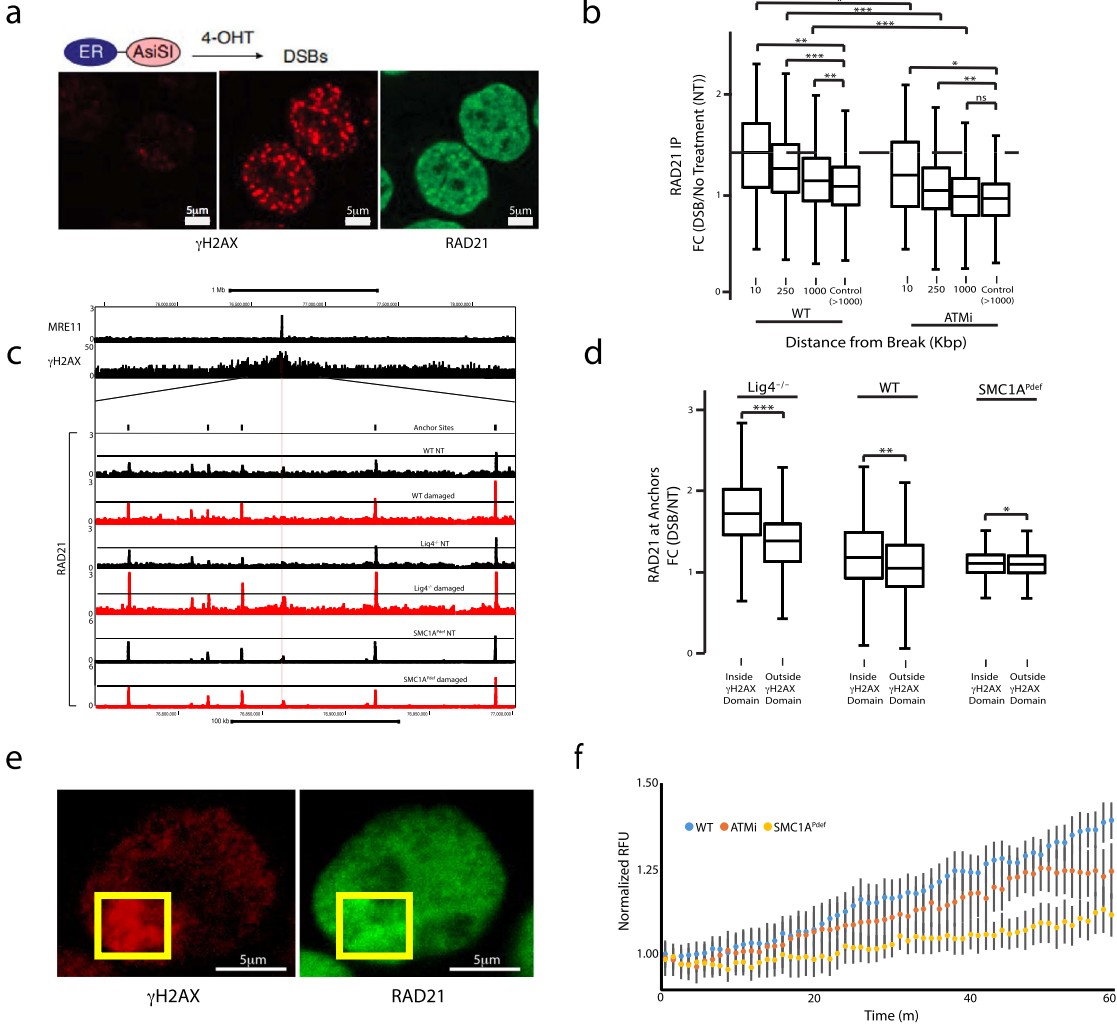

**Fig. 1 | Cohesin does not bind directly at DSBs, rather it accumulates in a phosphorylation dependent manner throughout damaged chromatin domains.** HCT116 cells containing inducible AsiSI were incubated with 4-hydroxytamoxifen (4OHT) to induce DNA damage. RAD21 ChIP-seq and HiC were performed in combination with MRE11 ChIP-seq to identify AsiSI DSBs. **a** Immunofluorescence (IF) staining 6 h after induction of AsiSI. The leftmost panel shows negative control untreated cells stained with γH2AX to compare with 4OHT treated cells in the middle (γH2AX-red) and right (RAD21-green) panels. RAD21 enrichment at γH2AX foci was observed. **b** ChIPseq was used to measure RAD21 enrichment at peaks around DSBs for WT cells ≤10 kb ($n = 73$, $P = 0.003$), ≤ 250 kb ($n = 765$, $P = 5E-11$), ≤1 Mb ($n = 1921$, $P = 0.01$) versus control chromatin >1 Mb ($n = 28,538$). ATM inhibited cells showed less significant loading around the break ($P = 0.04$, $P = 0.002$, and $P = 0.5$, respectively, for ≤ 10 kb, 250 kb, and 1 Mb vs. >1 Mb control). The number of asterisks assigned to a given item indicate the level of significance *$P < 0.05$, *$P < 0.01$ ***$P < 0.001$ (two sample, paired; T-test). Box plot parameters: The lower and upper limits of the boxes in the plot represent the 25th and the 75th quartiles, respectively. The central horizontal line represents the median, while each whiskers represent one interquartile range (IQR) defined by Q3-Q1. The data shown are from biological replicate. **c** Genomic view of a DNA DSB marked by MRE11 and γH2AX. The γH2AX domain is magnified to examine changes in RAD21 binding in damaged (red) and undamaged cells (black). Low-intensity binding sites not marked as anchors are NIPBL loading sites. **d** Cohesin anchor sites were determined by colocalization of RAD21 and CTCF. Enrichment of RAD21 peaks at anchor sites within damaged chromain domains was measured for WT ($n = 1467$, $P = 0.002$), LIG4[-/-] ($n = 1055$, $P < 1E-16$) SMC1A[Pdef] ($n = 1478$, P = 0.02) by comparing enrichment inside versus outside of γH2AX domains. Box plot and statistical parameters, as well as replicate information, are the same as panel **b**. **e** RAD21 (green) and γH2AX (red) observed in cells fixed 45 minutes post DSB induction with a UV laser. The yellow boxes indicate the area exposed to the UV laser. **f** Cohesin recruitment to DSBs was measured by live cell imaging over 1 hour for WT, ATMi treated, and SMC1A[Pdef] cells ($n = 25$ per condition). Error bars represent the Standard Deviation (SD) between treated cells. Source data for all panels are provided as a Source Data file.

---

when cohesin was depleted by over 95% in HCT-116 degron cells (Supplementary Fig. 2a), irrespective of damage conditions (Fig. 2). Additionally, we saw little gain or loss of loops associated with AsiSI mediated DNA damage (1 DSB specific loops gained and 2 lost among all AsiSI DSB sites), indicating that new loops do not form as a result of DNA damage (Supplementary Data 3). These findings argue that, despite significant cohesin accumulation, DNA damage has little or no effect on local chromatin topology.

Of note, addition of the ATM kinase inhibitor appeared to eliminate the slight increase in intra-TAD contacts following DNA damage

($P = 3E-5$, Fig. 2a). This could be attributed to the fact that ATM phosphorylates additional cohesin subunits[17], which may still function in SMC1A[Pdef] cells where the increase was observed. Alternatively, ATM inhibition could impact NIPBL-mediated cohesin loading. Further experiments will be necessary to fully investigate these possibilities.

In cohesin-depleted cells we observed a significant increase in intrachromosomal inter-domain contacts between damaged TADs relative to WT (FC = 1.16, $P = 1E-5$) as opposed to SMC1A[Pdef] or ATMi-treated cells, which showed only modest increases (fold change = 1.04 and 1.03, Fig. 2b, left panel). To a much lesser extent, a similar trend

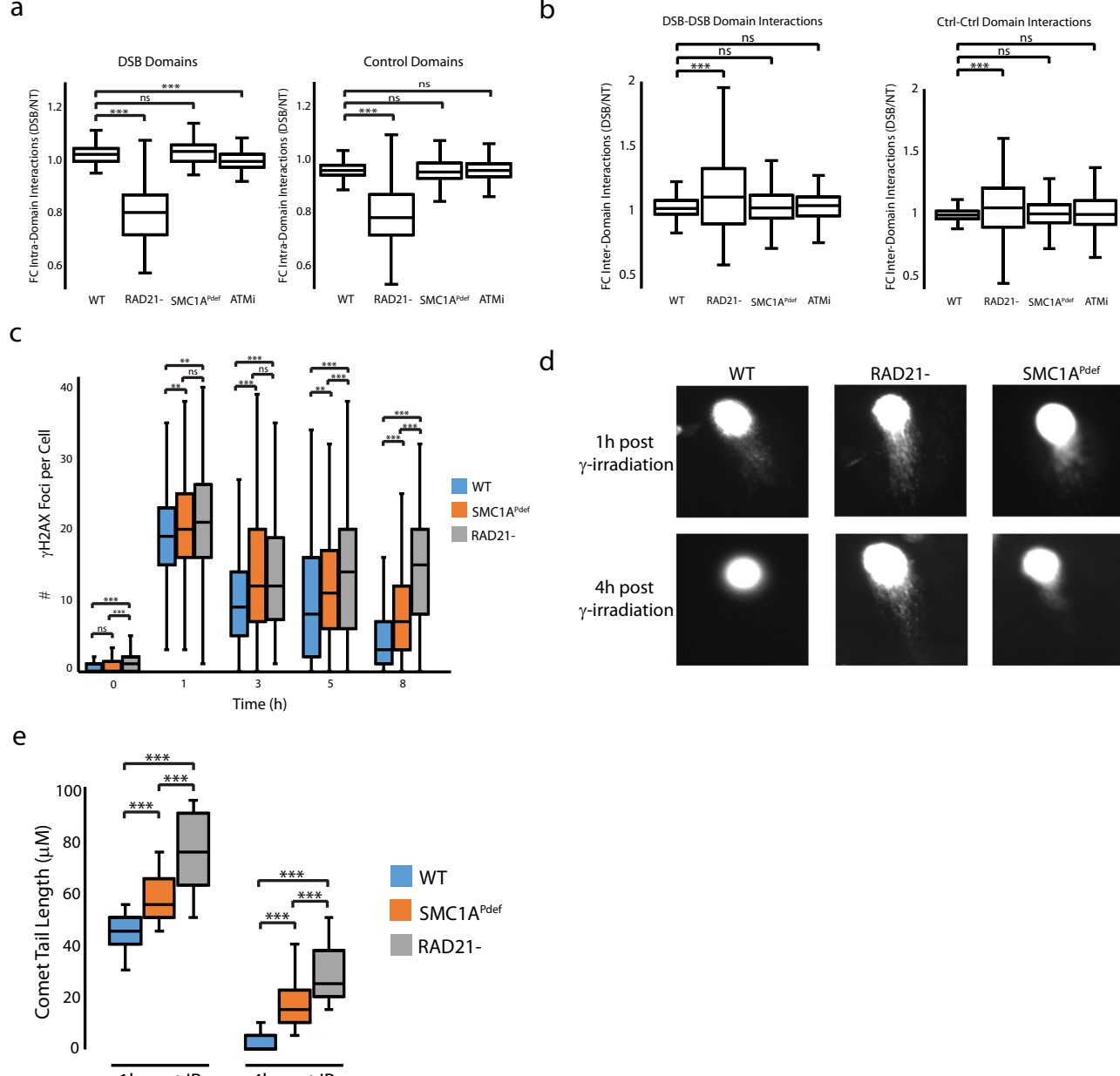

**Fig. 2 | Cohesin phosphorylation increases rate of repair but not contact frequency. a** Intra-domain interactions were measured for γH2AX domains and control domains not containing AsiSI breaks. Comparisons between damaged chromatin domains (*n* = 62) were made for WT versus RAD21- (*P* < 1E-16), SMC1A<sup>Pdef</sup> (*P* = 0.06), and ATMi (*P* = 3E-5). The same comparisons were made between undamaged control domains (*n* = 3425) for the same categories respectively (*P* < 1E-16, 0.06, 0.4) The number of asterisks assigned to a given item indicate the level of significance *P* < 0.05, **P* < 0.01 ***P* < 0.001 (one tailed, paired; T-test). Observed/ Expected (O/E) values were used to normalize experimental conditions. Data from two biological replicates was merged to produce this plot. Box plot parameters: The lower and upper limits of the boxes in the plot represent the 25th and the 75th quartiles respectively. The central horizontal line represents the median, while each whiskers represent one IQR. Outlying points are not shown. **b** Inter-domain interactions between intrachromosomal DSB-DSB domain pairs (*n* = 159, left panel) and Ctrl-Ctrl domain pairs (*n* = 159, right panel) were calculated between DSB/NT. DSB-DSB domains were compared between WT and RAD21- (*P* = 1E-6), SMC1A<sup>Pdef</sup> (*P* = 0.2), and ATMi (*P* = 0.7). The same comparisons were made for Ctrl-Ctrl domains respectively (*P* = 5E-4, 0.1, 0.08). Box plot, statistical parameters, O/E

normalization, and replicate information are the same as panel a. **c** HCT116 cells were treated with 2.5 Gy γ-irradiation and fixed at the timepoints designed to capture the entire DNA repair process. Immunofluorescence was performed for γH2AX and foci were imaged to measure the appearance and dissolution of foci. γH2AX foci were measured for WT, RAD21- and SMC1A<sup>Pdef</sup> cells at 0 h (*n* = 177, 174, 196), 1 h (*n* = 432, 358, 410), 3 h (*n* = 334, 260, 461), 5 h (*n* = 324, 291, 141), and 8 h (*n* = 304, 268, 269) post-treatment with 2.5 Gy γ-irradiation. Images were taken from three biological replicates and pooled. Box plot and statistical parameters are the same as described in panel a. **d** Representative images from comet assays performed under neutral conditions are shown at 1 h and 4 h post-treatment with 12.5 Gy γ-irradiation, showing delayed repair of DNA in RAD21- and SMC1A<sup>Pdef</sup> cells indicated by the migration of fragmented DNA (comet tail) at 4 h post-irradiation which is absent from WT. **e** Comet tail lengths were quantified from one of three biological replicate (n = 25 per condition). WT was compared to RAD21- and SMC1A<sup>Pef</sup> at 1 h (*P* = 1E-13, 2E-8) and 4 h (*P* = 1E-17, 2E-11) post γ-irradiation indicating defective repair. Replicates varied in intensity, but trends remained the same between backgrounds. Box plot parameter and statistical parameters are the same as panel a. Source data for all panels are provided as a Source Data file.

was noted for undamaged TADs in cohesin-depleted cells (fold change = 1.06 versus control, *P* = 0.001 Fig. 2b, right panel). This observation supports the model that damaged sites can aggregate in mammalian cells[30], but it is important to note that this activity is normally restricted by genome architecture. Overall, these data indicate that cohesin-mediated architecture prevents global DSB-DSB interactions.

We next explored whether cohesin impacts the recognition of DSBs by the repair machinery or directly impacts the efficiency of DNA repair. Since the continuous breaks generated by the AsiSI system hindered our ability to study repair kinetics, we employed γ-irradiation to introduce a controlled burst of DNA damage. We then measured repair rates at various time points. Specifically, we analyzed the accumulation kinetics of DNA repair foci, using histone H2AX phosphorylation as a proxy for DSB repair.

Employing a high-throughput imaging approach, we monitored the prevalence of γH2AX foci in cells with and without cohesin. Our results showed that the loss of cohesin only minimally increased the prevalence of γH2AX foci one-hour post-irradiation (FC = 1.10, *P* = 1E-4), suggesting that break sensing does not depend on cohesin (Fig. 2c). This increase is also consistent with enhanced chromosomal breakage associated with cohesin defects[42].

Previous studies proposed that γH2AX foci formation requires loop extrusion[10]. However, while we observed a 10-15% reduction in γH2AX signal intensity at various time points following cohesin depletion, foci remained detectable (Supplementary Fig. 2b, c). To further investigate this, we conducted ChIP-seq for γH2AX and confirmed the presence of γH2AX domains at DSBs (Supplementary Fig. 2d). Due to high background levels in HCT116 cells, we were unable to determine whether γH2AX domains spread as a result of cohesin depletion. Nonetheless, these findings indicate that cohesin is not essential for H2AX phosphorylation.

Despite seeing only a minimal impact on break sensing, we found that DSB repair—measured by the disappearance of γH2AX—was significantly delayed following cohesin removal (Fig. 2c). This delay was particularly pronounced at the 8-hour mark, where repair was complete in wild-type (WT) cells, while many foci persisted in RAD21-depleted cells (*P* < 1E-16, Fig. 2c). A similar delay in repair was also evident in phosphorylation-deficient SMC1A cells (*P* = 2E-11, SMC1AP-def vs. WT at 8 h).

To further confirm this delayed repair phenotype, we employed a neutral comet assay to directly measure DNA DSBs. Our results indicated that RAD21 and SMC1A^Pdef cells had significantly more damage 1 h post-γ-irradiation when compared to WT (*P* = 1E-13, 2E-8). Additionally, we found that repair was significantly delayed 4 h postirradiation in both RAD21-depleted (*P* < 1E-16) and SMC1APdef cells (*P* = 2E-11, Fig. 2d, e). This delay was replicated in mouse embryonic stem cells with RAD21 depletion using the same degron system[43] (Supplementary Fig. 2e). Additionally, we validated these findings by immunostaining for phosphorylation of the DNA repair factor KAP1, an ATM substrate[44] (Supplementary Fig. 2f).

Since cohesin modulates both short and long-range interactions[44] (Fig. 2a, b), we explored misrepair of cis chromosomal breaks in cohesin-depleted versus SMC1A phosphorylation-deficient cells. This comparison allowed us to evaluate whether delayed repair or chromatin architecture prevent end-joining errors. To this end, we designed qPCR primers to amplify deletions between adjacent AsiSI sites separated by 50kb-2Mb on the same chromosome (Fig. 3a and Table 1). The assay was optimized to generate single amplicons between 200 and 300 bp after induction of AsiSI damage, as illustrated in Supplementary Fig. 3a, b.

Under RAD21 depletion and continuous AsiSI DNA damage for 72 h, all six sites tested showed increased deletion junctions between AsiSI DSBs (Fig. 3b). This phenotype was easily distinguished by shifts in Cq values when cohesin was depleted (Supplementary Fig. 3c). Compared to WT cells, RAD21 depletion resulted in a six-fold increase

in cis-deletions (*P* = 2E-8, Fig. 3c). In contrast, SMC1A phosphorylation-deficient cells displayed a smaller increase in deletion frequency (FC = 2.1, *P* = 0.002), which was notably less than the levels observed with cohesin depletion (Fig. 3c).

Cohesin is essential for sister chromatid cohesion, and its depletion prevents cells from progressing through metaphase, leading to a buildup of cohesion-depleted cells in G2/M phase[45]. To mitigate this issue, we conducted experiments at confluence to reduce cellular turnover. Additionally, we utilized Palbociclib, a CDK4,6 inhibitor that arrests cells in G1, to ensure that cells remained in G1 during the 72-hour AsiSI treatment. Our results indicated that, upon induction of DSBs, the cis-deletion profile remained largely consistent between Palbociclib-treated and untreated cells (Supplementary Fig. 3d).

We next probed whether cohesin can prevent inter-chromosomal end joining events using the High Throughput Genome Translocation Sequencing (HTGTS) assay[46]. This technique employs a bait DSB to capture prey sequences, including cis-deletions and translocations, using a biotinylated bait primer, library preparation, and Illumina sequencing (Fig. 4a). As baits we selected isolated AsiSI-mediated DSBs, > 10 Mb from the nearest DSB, to maximize translocation signals and minimize proximity biases (Table 2). On average we measured 26% cis-deletions and 74%% translocations between our bait DSB and other AsiSI-induced DSBs, indicating that these baits were well suited for translocations analysis (Supplementary Fig. 4a). Notably, in all cases, over 90% of the captured translocations occurred between AsiSI break junctions.

Five baits were independently assessed to measure translocations after AsiSI DNA damage in the presence or absence of cohesin. When cohesin was depleted, translocations increased 4.3-fold between AsiSI DSBs compared to WT control (*P* < 1E-16, Fig. 4b and Supplementary Fig. 4b). This increase was consistent across all baits, as shown by K-means clustering after binning the data into 1Kb fragments (Fig. 4c). We also confirmed the results by performing qPCR for identified translocations (Supplementary Fig. 4c). In contrast, translocations in ATMi-treated cells only increased by half compared to the control (FC = 1.9, *P* < 1E-16), while SMC1A phosphorylation-deficient cells showed a similar increase (FC = 2.1, *P* < 1E-16; Supplementary Fig. 4b). These results support a model in which both the DNA repair and architectural functions of cohesin play critical roles in preventing misrepair of intra- and inter-chromosomal DNA breaks in mammalian cells.

We next probed the repair pathway responsible for the increased translocations observed in cohesin-depleted cells. Given that all our experiments were conducted under conditions that favor NHEJ, we performed HTGTS in the LIG4^-/- background. We found very few translocations after AsiSI induction in G1 inhibited cells (Fig. 4b). This is expected and distinctly different from the increase in translocations reported in G2 synchronized LIG4^-/- cells[47], which are thought to be mediated by HR. Importantly, cohesin depletion did not increase the number of translocations captured in the LIG4^-/- background (Fig. 4b and Supplementary Fig. 4b). This suggests that the translocations observed in WT cohesin-depleted cells are dependent on Ligase 4 and are thus a byproduct of NHEJ.

Next, we examined the use of microhomology in translocation events. We found that cohesin depletion did not alter the microhomology profile of WT translocations, indicating that the presence or absence of cohesin does not influence repair pathway preference (Supplementary Fig. 4d). The low levels of microhomology identified in WT cells further suggest that NHEJ is the primary repair pathway responsible for producing translocations under our experimental conditions. In contrast, the LIG4^-/- background served as a control, showing increased microhomology usage, as expected, since any repair events that occur must proceed through homology-based pathways (Supplementary Fig. 4d).

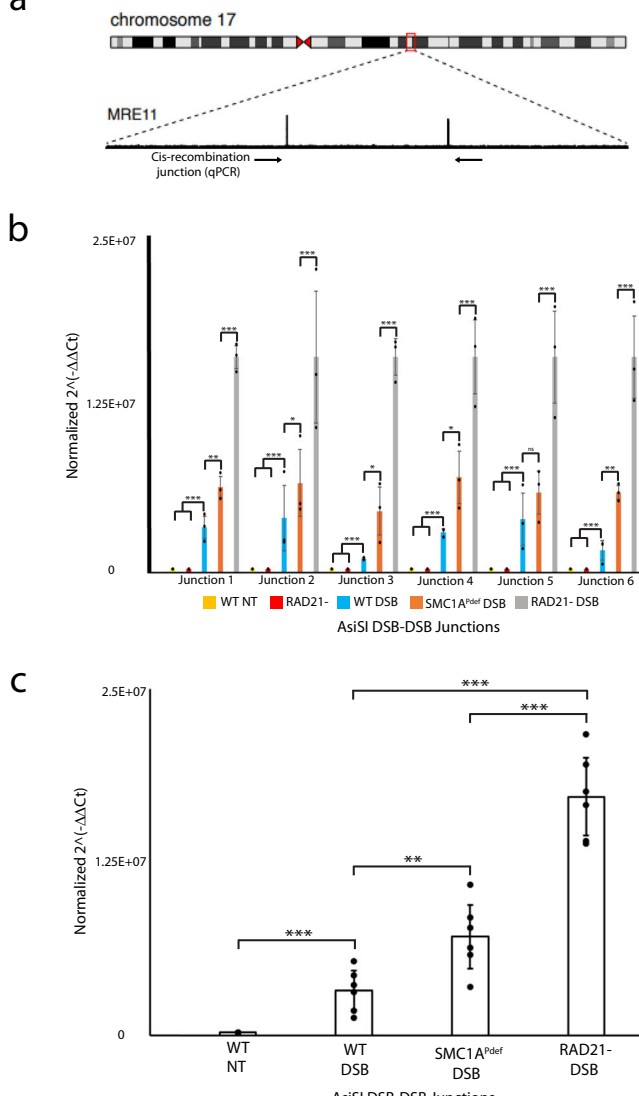

**Table 1 | Chromosomal positions of assayed DSB-DSB junctions**

| DSB-DSB Junctions | Chromosomal Position | Junction Distance (Kbp) |
|---|---|---|
| J1 | Chr22:20785436-20850310 | 68 |
| J2 | Chr09: 130693173-130889410 | 196 |
| J3 | Chr06:27145369-27661903 | 516 |
| J4 | Chr03:98618167-99536967 | 918 |
| J5 | Chr07:92157735-92861493 | 703 |
| J6 | Chr11:118965052-120196325 | 1231 |

those within γH2AX domains when cohesin was depleted (Supplementary Fig. 4f).

Additionally, we compared specific translocation events to contact frequency as measured by Hi-C and found increased interactions between translocated sites specific to AsiSI DSB induction in both WT (FC = 1.31, *P* < 1E-16) and RAD21- cells (FC = 1.39, *P* < 1E-16; Fig. 4d). Among damaged conditions, these enhanced inter-domain interactions were most pronounced at sites with high translocation frequency (Supplementary Fig. 4g). RAD21 depleted and SMC1A phosphorylation deficient cells exhibited increased interactions at translocated junctions when compared to WT (FC = 1.47, 1.31; *P* < 1E-16, P < 1E-16 respectively; Fig. 4d). This suggests that loss of cohesin phosphorylation increases inter-chromosomal contacts between damaged chromatin while cohesin depletion causes this effect globally for both short and long domains. This is consistent with the trend we observed in our analysis of DSBs (Fig. 2b). Additionally, from Fig. 4b it is clear that cohesin depletion leads to HTGTS sites translocating more broadly across the genome when compared to loss of phosphorylation. Thus, it appears that cohesin-mediated loops restrict DSB repair to specific chromosomal regions, while cohesin phosphorylation enhances the efficiency of end joining within these regions.

## Discussion

Our study presents evidence that cohesin plays two independent roles in DNA repair. First, cohesin's continual extrusion of chromatin maintains spatial separation between DNA DSBs, preventing cis-deletions and translocations. Second, ATM transient phosphorylation of cohesin in response to DNA damage accelerates the joining of DSBs.

While prior studies linked cohesin to DSB repair[15], it has been difficult to separate its loop extrusion function from its role as a repair factor. Furthermore, efforts to knock down cohesin have been hindered by the stability of cohesin binding and chromatin loops, which persist in partially depleted cells. To address these issues, we employed the AsiSI inducible system to create DSBs in cells either deficient in SMC1 phosphorylation or in cells where cohesin can be efficiently degraded.

To quantify DNA damage, we performed MRE11 and γH2AX ChIP-seq analyses. The results revealed that, despite strong MRE11 signals, cohesin is largely absent at DNA ends, instead accumulating at CTCF anchor sites within γH2AX+ TADs. This finding contradicts the hypothesis that DSBs serve as anchors for loop extrusion. Rather, cohesin's accumulation at damaged chromatin closely resembles the behavior of classical repair factors, which rely on ATM phosphorylation (e.g., SMC1) and accumulate considerably when DNA repair is compromised, such as in the absence of LIG4.

A key question is whether the localized increase in cohesin recruitment enhances loop extrusion. While previous studies reported a strengthening of TAD boundaries around DSBs[30], Hi-C measurements only showed a minor ~2% increase in intra-TAD contacts. This contrasts sharply with the substantial accumulation of cohesin observed through ChIP-seq (33%) and microscopy (38%). These

**Fig. 3 | Cohesin depletion and SMC1A^Pdef cells were compared to identify the proportion of cis-deletions derived from defects in genome architecture versus delayed repair. a** A representative example of cis-recombination measured by qPCR. Blue arrows indicate primers used to amplify the expected deletion. MRE11 ChIPseq peaks were referenced to identify consistent AsiSI DSBs pairs used for screening for deletion junctions. **b** HCT116 cells containing inducible AsiSI and RAD21-degron systems were induced for DNA damage and/or RAD21 depletion for 72 h. Six break pair junctions were measured by qPCR in the RAD21- and SMC1A^Pdef backgrounds. Dots represent biological replicates which were derived from the average of three technical replicates. Error bars represent SD between replicates. The number of asterisks assigned to a given item indicate the level of significance *P < 0.05, *P < 0.01 ***P < 0.001 (one tailed, paired; T-test). **c** All junctions were averaged to provide a global view of deletions between all baits in WT versus RAD21- (*P* = 2E-8) and SMC1A^Pdef (*P* = 0.002) cells. Each dot represents a junction sites (*n* = 6), while error bars represent SD between junction sites for a given condition. Statistical parameters are the same as panel **b**. Source data for all panels are provided as a Source Data file.

We found that under WT conditions, translocations primarily involved the break site or the damaged (γH2AX +) domain (Supplementary Fig. 4e). Notably, although the number of translocations increased in cohesin-depleted cells, their distribution between break sites and γH2AX+ domains remained unchanged (Supplementary Fig. 4e). However, we did observe a slight increase in the percentage of translocations occurring outside of γH2AX domains compared to

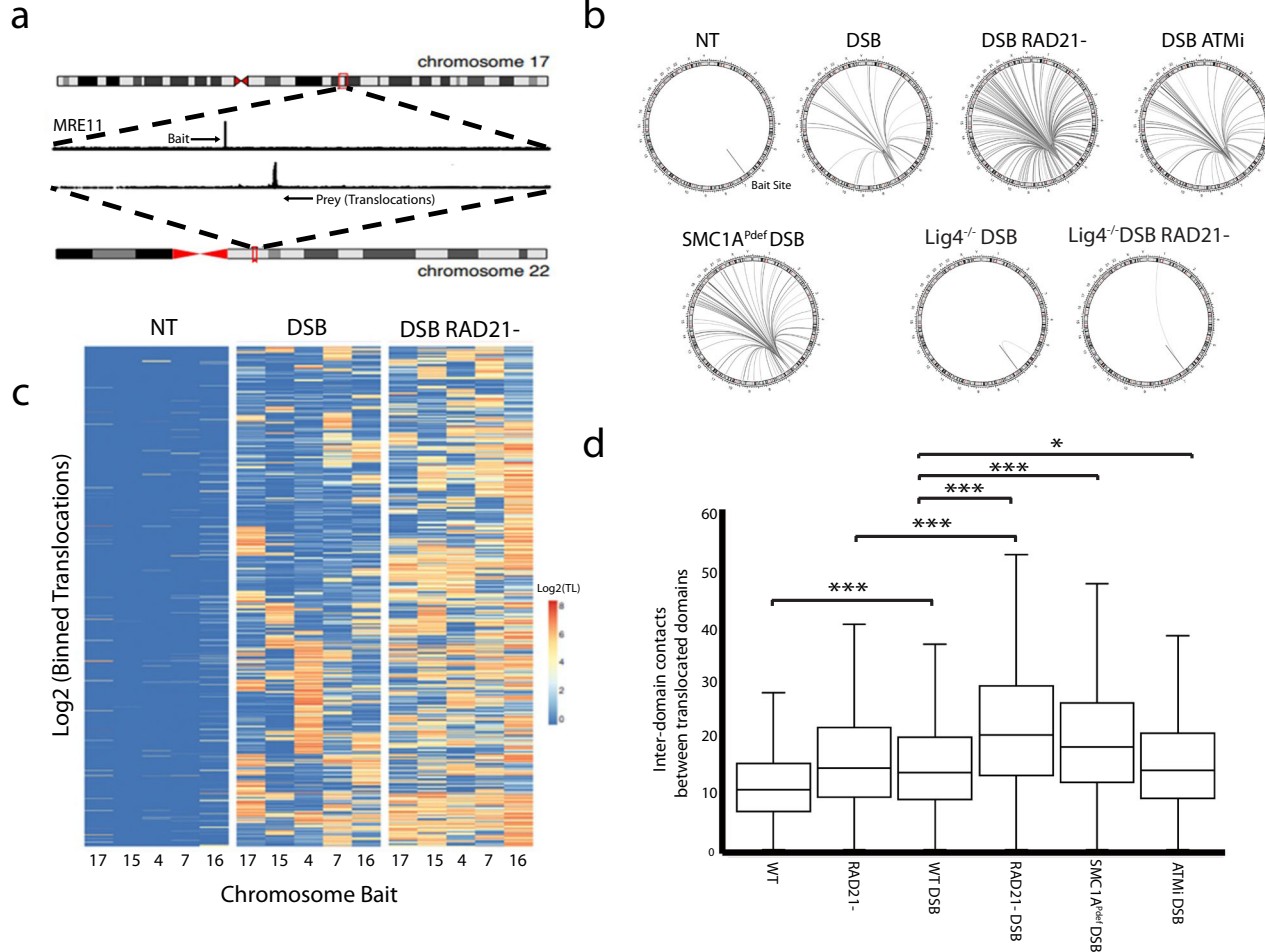

**Fig. 4 | The architectural and repair roles of cohesin function synergistically to promote error free repair during NHEJ.** HCT116 cells were induced for AsISI DNA damage in the presence or absence of RAD21 for 72 h prior to High Throughput Genome Translocation Sequencing (HTGTS). **a** An example translocation from a bait site (AsISI break) on Chr17 to a prey site on Chr22 (AsISI break) using HTGTS. **b** Circos plots were used to show the most prevalent translocation sites (≥ 600 unique events) for Bait 4 (chr7) for all experimental conditions. **c** Heatmaps were generated to illustrate the consistency of increased translocations produced by cohesin depletion across all bait sites. **d** Increased inter-chromosomal interactions between translocated sites (n = 2526) were observed for undamaged and damaged sites specific to DSB induction (P < 1E-16 WT v WTDSB, P < 1E-16 RAD21- v. RAD21-DSB). Among DSB-induced conditions, an increase in intensity was observed at translocated sites for RAD21-, SMC1A$^{Pdef}$, and ATMi treated cells versus WT (P < 1E-16, P < 1E-16, P = 0.05, respectively). The number of asterisks assigned to a given item indicate the level of significance *P < 0.05, *P < 0.01 ***P < 0.001 (one tailed, paired; T-test). Box plot parameters: The lower and upper limits of the boxes in the plot represent the 25th and the 75th quartiles respectively. The central line represents the median, while each whiskers represent one IQR. Source data for all panels are provided as a Source Data file.

findings prompted us to reconsider whether ATM-induced phosphorylation of cohesin primarily promotes repair via loop extrusion, as previously hypothesized, or if it operates through a different mechanism similar to classical repair proteins. In this alternative model, cohesin may associate with loop anchors through mass action, without necessarily driving chromatin extrusion.

Our results also established that γH2AX foci formation is not dependent on cohesin. This has been a source of confusion due to conflicting reports[10,28]. While we observed that the foci were slightly weaker and more diffuse in cohesin-depleted cells, the timing of H2AX phosphorylation remained largely unaffected. In contrast, DSB repair was significantly delayed both when cohesin was depleted and when it was not phosphorylated, highlighting cohesin's direct role in accelerating the repair process.

We observed that cis-deletions and chromosomal translocations occur frequently when cohesin is depleted. Furthermore, experiments in LIG4$^{-/-}$ cells revealed that both events are mediated by the NHEJ repair pathway. Interestingly, while the number of translocations increases in SMC1A$^{Pdef}$ mutant and ATMi-treated cells, the increase is significantly less pronounced than in the absence of cohesin. These findings suggest that cohesin's canonical role in TAD formation helps prevent damaged chromatin domains from interacting, thereby contributing to genome stability beyond its function as a repair factor. The elevated translocations in SMC1A$^{Pdef}$ cells likely result from delayed repair, giving damaged chromatin more time to interact.

There are at least two potential mechanisms through which cohesin phosphorylation could facilitate end joining. First, phosphorylation may pause loop extrusion to prevent interference with DSB processing and ligation. This idea aligns with the proposal that the repair machinery halts cohesin progression in mammalian cells[48]. An alternative possibility is that phosphorylation enhances cohesin's interaction with, and recruitment of, DNA repair factors directly involved in DSB end joining. Further research will be required to elucidate the precise mechanisms underlying these processes.

**Table 2 | Chromosomal positions of assayed HTGTS baits sites**

| AsiSI Bait Site | Chromosomal Position |
|---|---|
| Bait 1 | Chr17: 18057102 |
| Bait 2 | Chr15: 59225292 |
| Bait 3 | Chr04: 178363578 |
| Bait 4 | Chr07: 29604609 |
| Bait 5 | Chr16: 56966412 |

## Methods

### HCT-116 degron system
We obtained HCT-116-CMV-OsTir1 and HCT-116-RAD21-mAID-mClover cells (HCT-116 RAD21-mAC) from PMID:27052166. Cells were cultured using McCoy's 5 A media with 10% FBS, 2 mM L-glutamine, 100 U/mL penicillin, and 100μg/mL streptomycin. Cells were grown at 37 °C and 5% $CO_2$ in a humidified incubator. HCT116-RAD21-mAID-clover cells were grown for 3 days before addition of media containing 750μM indole-3-acetic acid (IAA, auxin) (Millipore Sigma, I5148). A complete list of materials along with the corresponding product identifiers can be found in Supplementary Table 1. RAD21 is completely degraded after 6 hours IAA treatment and was routinely monitored using flow cytometry. Maintaining RAD21 degradation over long time courses can be achieved by changing auxin media every 24 h. All cell lines tested negative for mycoplasma. No commonly misidentified cell lines were used in the study.

### CRISPR Cas9 engineering of HCT116
The LIG4[-/-] mutant was obtained in HCT116-RAD21-mAID-clover background using CRISPR Cas9. Suitable sgRNA targets were identified using the sgRNA online tool https://crispr.zhaopage.com, and sgRNAs were cloned into the pX330-BSD vector. To delete a target gene, HCT cells were transfected using the Cell Line Nucleofector™ Kit V (Lonza, VCA-1003) with the Nucleofector™ 2b device. For each target gene, 2 μg px330-BSD containing an sgRNA targeting a sequence upstream the first exon and 2 μg px330-BSD containing sgRNA targeting a sequence downstream of the last exon were used (Supplementary Table 2). Forty-eight hours following transfection, cells were treated with Blasticidin at a concentration of 5μg/ml. After 7-10 days of selection, cells were diluted into 96 wells at a concentration of 1 cell per well. Genotyping was performed using WT primers to confirm the absence of a band and primers specific to the deletion of interest to confirm the presence of a band (Supplementary Table 2). Cell lines were validated by western blot using a monoclonal antibody against DNA Ligase 4 (Cell Signaling Technologies, 14649S). A complete list of antibodies along with the corresponding product identifiers can be found in Supplementary Table 3.

To generate homozygous SMC1A[S957AHz/S966AHz] referred to as SMC1A[Pdef] a single sgRNA was used to generate DSBs in the exon of interest. HCT cells were transfected concurrently with a homology repair (HR) construct containing two, 500 bp homology arms flanking a modified sequence on both sides. The modified sequence contained the appropriate Serine to Alanine mutations at positions 957 and 966 as well as silent mutations throughout a 60 bp region surrounding these mutations. Primers were designed against the modified and WT sequence to perform genotyping after selection and single cell dilution (Supplementary Table 2). Cell lines were validated by western blot using an anti-SMC1A (phospho S957) antibody (Abcam, ab137871).

### Vector cloning and transduction
We obtained the HA-ER-AsiSI retroviral vector, which was modified using stich PCR to generate the HA-ER-AsiSI-T2A-BFP retroviral vector used in this study. Retroviral particles were produced in the Platinum-A retroviral packaging cell line by transfecting ER-AsiSI-T2A-BFP in the presence of Lipofectamine™ LTX Reagent with PLUS™ Reagent (ThermoFisher Scientific, A12621). Infectious retrovirus was harvested 48 hours post-transfection. WT, 53BP1[-/-], and LIG4[-/-] cells were transduced with retrovirus ER-AsiSI-T2A-BFP by centrifugation for 90 min at 1000x *g*, at 25 °C in the presence of 4 mg/ml polybrene. After 4 days, the cells were FACs sorted in bulk for BFP expression and single-celled by dilution. Single colonies were screened using IF microscopy and selected for based on the following criteria: 1) Low background γH2AX foci in untreated cells 2) Distinct and uniform γH2AX foci in DSB induced cells 3) Uniform BFP expression 4) Uniform RAD21 depletion.

### AsiSI Induction and ATM inhibition
HCT-116 cells containing the nAsiSI-T2A-BFP construct show efficient induction of breaks 4-8 hours following treatment with 1μM 4-Hydroxytamoxifen (4OHT) (Millipore Sigma, SML1666). In experiments where DNA damage was induced for 72 h, fresh 4OHT was added every 24 h. Inhibition of ATM was achieved by incubating cells with 1μM KU-55933 (Fisher Scientific, 35-441-0) for 2 hour prior to induction of DSBs.

### Live cell imaging of cohesin recruitment
HCT116-RAD21-mAID-clover cells were plated onto an Ibidi μ-dish 4 days. Prior to imaging, cells were incubated with 1.5μM Hoechst 33342 (ThermoFisher Scientific, 62249) for 1 h. Following incubation, the cells are washed, and the media is replaced with FluoroBrite DMEM + 10% FBS. The μ-dish was then fitted with a glass coverslip and placed into the incubation (37 °C and 5% $CO_2$) stage of a Zeiss LSM 880 NLO Laser Scanning Microscope with 40x water objective. DNA breaks are induced by using a 355 nm UV laser (10% power) to damage the designated section of the nucleus, producing an average of 1 break per Mb. RAD21-clover recruitment is then measured using a 488 nM argon laser and imaged (12 scan average) at intervals of 1 min. Zen Black software was used to correct photobleaching and background noise. Definite focus was used on 1-minute intervals to minimize drift and maximize signal intensity.

### Immunofluorescence staining
Cells were grown in ibitreat μ-slides (Ibidi, 80826) and fixed in a solution of 4% Paraformaldehyde (Electron Microscopy Sciences, 15710) for 10 min at room temperature and washed with cold PBS three times. Cells were permeabilized with 0.25% TritonX-100 for 10 min. Slides were washed with PBS three times before being resuspended in a blocking solution containing PBS-1%BSA-0.05%Tween20 overnight. γH2AX-A555 antibody (Millipore Sigma, 05-636-AF555) was added to fresh blocking solution at a concentration of 1/2500 and incubated with cells for 1 hour at room temperature. Following incubation, cells were washed 3 times with PBS-0.05%Tween20. For nuclei staining, cells were treated 300 nM DAPI (ThermoFisher Scientific, 62248) for 5 min before resuspending in PBS or mounting media for imaging.

### Flow cytometry
Live cells were resuspended in indicator-free McCoy's 5 A and 10%FBS. Cells were analyzed using the BD FACSCanto or sorted using the BD FACSAria IIIu. Live/dead cell stain was used for analysis and/or sorting of live cells. In the case of fixed cells, HCTs were trypsinized and resuspended in 4% PFA for 10 min and washed 3 times with cold PBS. For cell cycle analysis, HCT cells were treated with 300 nM DAPI for 5 min before resuspending in PBS + 2% FBS. When performing assays related to Kap1, cells were treated with phospho Kap1 antibody at a 1/2000 dilution (Bethyl Laboratories, A300-767A) in 1x BD Perm/Wash Buffer (BD Bioscience, 554723).

### γ-irradiation and γH2AX foci analysis
HCT cells grown on μ-slides were treated with either 2.5 Gy γ-irradiation. After exposure, cells were returned to incubators before being fixed at different time points. IF was performed and cells were

imaged with a 40x objective. Foci were automatically counted using the BioTek Gen5 Software for Imaging and Microscopy https://www.agilent.com/en/product/cell-analysis/cell-imaging-microscopy/cell-imaging-microscopy-software. The program first identifies nuclei based on DAPI signal (primary mask ≥ 5,000 RFU, size = 10μM-20μM. Nuclei were then analyzed with the A555 channel to identify the number of foci per nucleus (secondary mask ≥ 5,000 RFU spot size = 0.2μM-2μM). Thresholds were established for foci intensity using untreated cells as a negative benchmark and LIG4$^{-/-}$ as a positive benchmark. Replicates were imaged and stained on the same day with identical laser configuration and analytic parameters. Following analysis, each image was checked manually for errors in processing.

## Comet assay

We performed the neutral comet assay following the protocol outlined by the Comet Assay Kit (Abcam, ab238544). Briefly, treated cells were combined with low melting temp agrose at 37ºC to a final concentration of 5×10$^4$ cells/mL. This agarose-cell mixture was then plated on microscopy slides containing a base layer of agarose. Once cooled, microscopy slides were submerged in lysis buffer for 12 h. Slides were then washed in TBE and transferred to an electrophoresis chamber. Electrophoresis was performed at 20 volts for 25 minutes under neutral conditions (TBE). Following electrophoresis, slides were washed in EtOH and stained with Vista Green DNA dye. Slides were dried until agarose was flat and imaged. Due to the sensitivity limits of this assay, a dose of 12.5 Gy of γ-irradiation was used to achieve consistent and measurable DNA fragmentation.

## Hi-C

Cultured cells were fixed with 4% formaldehyde and the reaction was quenched with 200 mM glycine. Cells were then lysed and digested with the MboI restriction enzyme. DNA ends were then marked with biotin-14-dATP and ligated by proximity. After reverse crosslinking, DNA was sheered to an average size of 200-400 bp. This DNA was then pulled down using Dynabeads™ MyOne™ Streptavidin T1 (ThermoFisher Scientific, 65601) and washed. Hi-C library were amplified directly from of the beads with 4-6 cycles of PCR, using Illumina primers (Bioo Scientific) and Q5® High-Fidelity DNA Polymerase (New England Biolabs, M0491S). Libraries were sequenced with 100 bp paired-end sequencing using the NovaSeq6000 Illumina.

## Translocation and Cis-Deletion qPCR

Cells pellets were incubated with DNA lysis buffer (200 mM NaCl, 10 mM Tris-Cl pH8.0, 10 mM EDTA, 0.5% SDS, and Proteinase K (0.1 mg/mL final concentration)) for 24 h at 55°C. Tubes were spun down and the pelleted DNA was washed with ethanol before being eluted in 200uL TE buffer. Following DNA extraction, PCR reactions were prepared using PowerUp SYBR Green Master Mix (ThermoFisher Scientific, A25780), Forward Primer (10μM), WT or deletion reverse primer (10μM), and 10 ng gDNA template. PCR reactions were optimized using gel electrophoresis and melting temperature analysis to ensure primer specificity for single products. Forward primers were always designed to be universal for both WT and deletion PCR products. The reverse primers vary in position to amplify the WT configuration or the deletion configuration. All amplicons were designed to be between 150-250 bp in length.

## Chromatin immunoprecipitation sequencing (ChIP-seq)

Cultured cells were fixed with 4% formaldehyde for 10 min at room temperature and the reaction was quenched with 125 mM glycine. Twenty million fixed cells were washed with PBS, snap-frozen and stored at −80 °C until further processing. Before use, the cells were resuspended in 1 ml of RIPA buffer (10 mM Tris pH 7.6, 1 mM EDTA, 0.1% SDS, 0.1% sodium deoxycholate, 1% Triton X-100 and freshly added cOmplete™, Mini, EDTA-free Protease Inhibitor Cocktail

(Millipore Sigma, 11836170001). Sonication was performed using Branson sonicator at an amplitude 35%, 12 cycles of 20 sec sonication followed by 30 sec of pause or using Covaris ME220 sonicator at duty factor 20%, peak incident power 75, cycle/burst 1000 for 6 min at 4°C. For each RAD21 ChIP, 5μL antibody (Abcam, Ab992) was mixed with 100μL PBS. For each MRE11 ChIP, 4μL antibody (Novus Biologicals, NB100) was mixed with 100μL PBS. For each gH2AX ChIP, 5μL antibody (Abcam, Ab81299) was mixed with 100μL PBS. The antibody-protein solution was then mixed with (40 μl initial volume) of Dynabeads™ Protein A (ThermoFisher Scientific, 10002D), which have been drained of the liquid volume by magnetic separation. This solution is suspended by pipetting and incubated at 40 min at room temperature under agitation. Antibody-bound beads were washed twice with RIPA buffer, twice with RIPA buffer containing 0.3 M NaCl, twice with LiCl buffer (0.25 M LiCl, 0.5% Igepal-630, 0.5% sodium deoxycholate), once with LiCl buffer (0.25 M LiCl, 0.5% NP-40, 0.5% NaDOC), once with TE pH 8.0 containing 0.2% Triton X-100, and once with TE pH 8.0. Crosslinks were reversed by incubating the beads at 65 °C for 4 hours in the presence of 0.3% SDS and 1 mg/ml Proteinase K. ChIP DNA was purified by ChIP DNA clean and concentrator column. Libraries were prepared using the Ovation® Ultralow System V2 (Tecan, M01379) and 50 bp single-end sequencing was performed on HiSeq3000 or NovaSeq. The data shown in this manuscript are derived from WT and LIG4$^{-/-}$ clones denoted D8-WT and A1-LIG4. To ensure reproducibility we utilized different clones denoted C5-WT and A12-LIG4 for our second replicate. Our submission also includes two replicates of the SMC1A$^{Pdef}$ clone denoted P2A11_RA4. All RAD21 IPs were done in parallel with MRE11 ChIPseqs. For analyses using MRE11 as an indicator for DNA DSBs, we used condition/background-specific MRE11 IPs for quantification.

## HTGTS

HTGTS was performed as previously described[46]. Briefly, 20 μg genomic DNA from HCT-116 cells with indicated genotype were sonicated and subjected to LAM-PCR. Five AsiSI cutting sites were selected as bait DSB for junction amplification. After streptavidin magnetic beads enrichment and adaptor ligation, the ssDNA fragments are amplified with barcode primers for sequencing. Sequencing was performed using the Illumina HiSeq. Oligo sequences are listed in Supplementary Table 2. The raw Hiseq data were passed through several quality control steps as previously described[46]. Processed reads were mapped to hg19 genome by Bowtie2 through the HTGTS pipeline with "--no-dedup" parameters (https://github.com/robinmeyers/transloc_pipeline). HTGTS junctions in ±20 kb from the bait site were annotated as "bait site" and removed from translocation analysis. Intra- and inter-chromosomal translocations were calculated with Bedtools. The Metagene profile was analyzed and plotted with Deeptools. SMC1A$^{Pdef}$ cells were processed at a later time. To prevent batch effect variation, these experiments were processed concurrently with WT baits for normalization.

## ChIPseq Analysis

Reads were sequenced using the illumina HiSeq 3000 or NovaSeq 6000 following the manufacturer's instructions. Raw reads were aligned to human (hg19) genome using bowtie1 with flags *-S -m 1 -a --best --strata -n 2*, and aligned reads were selected with *samtools view -S -b -F4* and sorted[49]. PCR duplicates were removed using Picard, and multiple aligned reads were removed using Samtools[50]. Reads are extended into the estimated fragment size by MaSc program. Density tracks were generated using custom software based on the samtools library to count the number of reads in 100 bp windows normalized to window size and library size to obtain densities in units of reads per kb region per million mapped reads (rpkm) across the genome and converted to bigWig with bedGraphToBigWig (UCSC utilities). All RAD21, CTCF and Nipbl de-duplicated ChIP-seq alignment files and input

alignment files are provided in MACS2 for the peak calling with -q 0.001 options. Each count is added 0.1 pseudo counts and then normalized with the number of mapped reads after deduplication. All plotting was done with the R package ggplot2. Methylation at AsiSI sites was determined using bisulfite sequencing data from HCT116[51]. Peaks calls are reported for all samples with MRE11 (Supplementary Data 2) and RAD21 (Supplementary Data 3).

## Hi-C Analysis
Using Juicer software[52],.hic files were generated, and normalized ligation frequency matrices were obtained with a dump command. Interaction matrices were visualized by Juicebox software. 4C-like track around DSBs are generated by calculating log2 fold change of interactions w/wo DSBs from the row region within +/- 2 Mb around DSB containing bin in normalized ligation frequency matrices. To avoid infinite value pseudo count 1 is added for log2FC calculation. For interchromosomal interaction matrices, we extracted normalized matrices on chromosomes, which have translocations in 250 kb resolution to save the space and time for matrix calculation. Interchromosomal domain Interactions between DSBs were calculated by summing up the surrounding 2 bins of each side ofthe translocating interaction point containing bin (1.25MB*1.25MB region). To calculate log2 fold change without infinite, we added pseudo count 0.1 to the summed interaction.

## Statistical tests
For comparisons of two groups with equal n values, one tailed, paired T-test was used. For comparisons between groups with different n values, a two sample, paired; T-test was used. Significance values for all figures are denoted: * $P = 0.05$ ** $P = 0.01$ *** $P = 0.001$.

## Reporting summary
Further information on research design is available in the Nature Portfolio Reporting Summary linked to this article.

# Data availability
ChIPseq and HiC data generated in this study have been deposited in the Geo database under accession code GSE250510. The processed ChIPseq and HiC data are provided in the Supplementary Information/ Source Data file. The HTGTS data used in this study are available in the SRA database under accession code PRJNA1087267. Source data are provided with this paper.

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

## Acknowledgements
We would like to thank lab members Jens Kalchsmidt, Pawel Trzaskoma, and Daniel Chauss for training and materials support. We would like to thank NIAMS Flow Cytometry Core members Kevin Tinsley and James Simone as well as Stephania Dell'Orso from the NIAMS Genomic Technology Core. Chinese Academy of Sciences Grant, 318GJHZ2022010MI (FM). National Insititues of Arthritis and Musculoskeletal and Skin Diseases Intramural Research Program, 1 ZIA ARO41148-20 (MF).

## Author contributions
Y.S. performed HTGTS and analysis. S.J. acted as primary bioinfomtician for ChIPseq, HiC, and all general analyses. K.F. performed downstream bioinformatic analysis of HTGTS, ChIPseq, and HiC data. S.P. performed several ChIPseq exeriments and one HiC replicate. D.M. advised with live cell imaging analysis. R.S. performed cell cycle experiments, valadiation experiments, and γH2AX ChIPseq. H.N. performed replicates of HiC. D.Z. advised and helped analyze foci experiments. H.T. analyzed HTGTS data and gave statistical guidance. S.J. performed validation experiments. H.F. advised and helped anlayze COMET experiments. A.C. peformed cell viability assays. S.V. performed mES imaging. JW provided imaging expertise and analysis for all imaging experiments. V.S., J.O., L.E., A.N., M.A., and F.M. provided specialized guidance and training, resources, and revisions. MF wrote the manuscript, performed all genome editing, and performed all other experiments. R.C. oversaw and guided M.F. in all aspects of this project project.

## Funding

## Competing interests
The authors declare no competing interests.
