## [Transparent Peer Review file · Nature Communications]

A Dual Role of Cohesin in DNA DSB Repair

Corresponding Author: Dr Michael Fedkenheuer

Version 0:

Reviewer comments:

Reviewer #1

(Remarks to the Author)

In this manuscript by Casellas and collaborators, the Authors analyze the role of cohesins in the DNA damage response and in the repair of DNA double strand breaks. They set out to address how cohesins may control chromatin architecture and chromatin loop extrusions upon DNA damage, how cohesins may control the DNA damage response and the spreading of gH2Ax domains intra and inter-TADs, and the role of cohesins in DNA repair and genome instability. They took advantage of the ER-AsIS system to introduce programmed double strand breaks in cell lines engineered with either a degradable RAD21-AID knock-in (for loss of function studies) or a mutated SMC1A (SMC1A-Pdef, lacking ATM dependent phosphorylation site), thus enabling the researchers to potentially sort out cohesins activities controlled by ATM phosphorylation.

The Authors show that Rad21 does not localize at DSBs, yet DSBs stimulate Rad21 binding to damaged regions (TADs), and within these regions Rad21 accumulates more prominently at anchor sites decorated by CTCF. Yet, increased cohesins at damaged TADs has a marginal effect on intra-TAD contacts (only a slight increase is reported). Based on the lack of localization of cohesins at DSBs and lack of a strong increase in intra-TAD contacts the Authors exclude that cohesins may promote loop extrusion at double strand breaks. Also, the spreading of gH2Ax domain was independent of cohesin. They also provide evidence that cohesins are needed to prevent long-distance chromosomal contacts and promote conservative DNA repair (and prevent chromosomal translocations). This latter function depends on Pi of SMCA1.

Importantly, these data are in stark contrast to what was recently proposed by Arnould et. al, (Nature 2021) where cohesins were shown to orchestrate the remodelling of DNA damaged TADs by (i) binding DNA-broken ends present at DSBs, (ii) by promoting loop extrusion, (iii) driving gH2Ax spreading (intra-TAD) and (iii) limiting gH2Ax spreading to adjacent TADs. All these findings and the underlying model are strongly challenged by the work of Casellas, which fails to confirm the main data.

Also, the data in the Casellas' manuscript suggests a role for cohesins in preventing chromosomal translocation, a conclusion that seems to put into question what reported by Arnould and collaborators (Nature 2023), that showed that loss of cohesins reduced non-conservative rejoining of DNA breaks (thus decreasing chromosomal translocations). These inconsistencies are perplexing and not easily explained considering that the Casellas group used cellular models and experimental strategies that are similar to what was previously employed by Arnould et. Al, (Nature 2021 and 2023). If confirmed, the Casellas data would raise serious concerns on the previously published works.

General points

The manuscript needs to be improved in terms of statistical robustness in order to be able to make a strong case, thus replicates are needed for all NGS analyses. Also, considering that this work strongly challenges previous work, re-analysis of published data is suggested.

Major points:

1. All ChIP-seq need replicates and possibly the use of spike in (as used in Arnaud et. Al, Nature 2021). At present, it appears that analyses were performed on a single sample for each condition (based on the data deposited in GEO by the Authors).
2. Authors need to provide evidence for the consistency of DSBs detected in the different cellular models and among replicates. In particular, they need to run replicates of the MER11 ChIPseq in WT cells and SMCA1Pdef cells. Also, since many of the analyses rely on the assumption that DSBs identified on WT cells will also be conserved in Lig4^{-/-} cells, it is also needed to perform the MRE11 ChIPseq in Lig4^{-/-} cells (in duplicates).

2. Please provide, as supplemental data (excel file) or as processed data in GEO (bed file), a list with the genomic coordinates of ChIPseq peaks called in all the replicates.

Minor points

1. Fig1.B (and all other figures): when reporting results of the statistical tests, please indicate the groups/conditions evaluated.
2. Fig1b: are the reported Rad21 signals calculated from Rad21 peaks or they are the average Rad21 signal in the TAD?
3. Fig1b,d: please provide a ranked heatmap of the Rad21 signals for all the groups reported in the box plot.
4. Line 107: the statement is supported by the observation of a single region, thus hardly generalized, can the Authors extend this analysis to all the DSBs?
5. Line 119: Auth. state that accumulation of Rad21 at anchor sites depends on SMC1A phosphorylation, yet it is unclear whether the DSBs found in WT cells are also found in SMC1A Pdef cells. Please provide evidence in support of DSBs consistencies in the different cellular models used.
6. Fig 1F: Rad21 recruitment to damaged DNA is slow compared to what commonly observed with other proteins recruited to DNA breaks. Do the Authors have an explanation? Can they provide some additional positive (a GFP-labelled DDR protein, which are usually recruited fast, or GFP labelled-repair factors, which usually show delayed recruitment) and also a negative control (GFP alone)?
7. Fig 2a and b: while the analysis shows differences of the average of the distributions, this analysis does not show whether, for each gH2Ax domain considered, there is indeed (i) a decrease of intra-domain contacts and (ii) an increase of inter-domain contacts. This is especially relevant for the analysis displayed in fig 2b where, given the high dispersion of the Rad21 -/- data, there could be a relevant fraction of gH2Ax domains in which interdomain interactions are instead decreasing. Thus, the Authors should complement these analyses with evidence of the fraction of gH2Ax regions that show decrease of intra-domain interactions and increase of inter-domain interactions upon Rad21 loss.

Reviewer #2

(Remarks to the Author)

It has been known for over two decades that SMC1A and SMC3 are substrates of the ATM (and ATR) protein kinase and are phosphorylated on defined serines in response to DNA damage. Mutations of the target sites in SMC1A results in abnormal responses to DNA breakage, including abnormalities in cell cycle checkpoints and enhanced chromosomal breakage after ionizing irradiation. It has remained unclear, however, how this DDR role of SMC1A/SMC3 (cohesin) is related to the known major role of cohesin in regulating higher order chromatin interactions. In this submission, Fedkenheuer et al. perform a series of experiments to try to address this gap in understanding. They provide evidence for a dual role of the cohesin complex in helping cells deal with DNA breakage, one related to ATM/ATR phosphorylation of SMC1A and a second role in insulating damaged DNA/chromatin from long-range interactions, thus influencing the formation of translocations and intra-chromosomal deletions – the latter insight represents the major advance from this body of work. Some clever experimental approaches are used to separate out these functional roles and the insights gained represent a novel addition to the literature. However, the mechanistic insights gained are still somewhat modest.

Specific Comments:

1. Some nice approaches are used to address specific questions, such as the AsiSI system to introduce defined sites of DNA breaks and combining this system with Lig4-deficient cells to prevent re-cutting, which has been a challenging issue with inducible nuclease systems like this. In addition, the application of HiC and HTGTS technologies to these questions represent an important approach and enabled the bottom-line conclusions.
2. Several of the results confirmed insights that were already in the literature. The main novel insight relates to the HTGTS results shown in Fig. 4. Given the importance of the results of this assay and the relatively small quantitative effects of some of their modulations in other assays, they should better quantitate the differences in translocations under the different conditions, including adding the ATMi and SMC1APDef curves to Fig. 4D.
3. They relied to a certain extent on disappearance of gH2AX foci as a surrogate marker for DNA repair. While this is a commonly used approach in the literature, it is somewhat problematic in that it is only a surrogate for repair of DNA breaks and can be misleading; for example, inhibition of gH2AX de-phosphorylation would result in failure of disappearance of the foci but is not representative a failure to repair (examples of this happening are in the literature). To the authors' credit, they acknowledge that foci disappearance is only a surrogate and address this with the comet assay in Fig. 2D/E. Since they have the AsiSI system in place, they have the capability of directly measuring DNA break repair as re-ligation of the broken sites; it would be elegant to also use this approach directly measure DNA break repair if possible.
4. Many of the effects are quantitatively quite small; they might be statistically significant, but one wonders about the biologic significance of some of these differences. Some comments from the authors on these particular issues would be appreciated. The good news is that that translocation effects are fairly convincing.
5. One aspect of their proposed model of how SMC1A/cohesin is involved in DNA breakage responses that remains unclear is whether SMC1A/cohesin is being recruited to the break sites or whether cohesin complexes that are already in place near where breaks occur are affecting long-range interactions. If the former, this appears to not require ATM or SMC1A phosphorylation. Could the authors provide some clarity on their thinking in this regard (both what may be concluded or what remains to be elucidated)?
6. Could the authors comment on how these insights link to the published data showing that SMC1A phosphomutants exhibit significantly increased chromosomal breakage after irradiation, similar that seen in ATM-deficient cells? Are these observations consistent with their data?
7. Data in Fig. 2 suggests that ATM kinase inhibition reduces intra-domain contacts, but that this effect is independent of

SMC1A phosphorylation and independent of DNA damage. This observation seems confusing relative to their model and requires some explanation.

8. The last sentence in the discussion states that “.....inhibitors of cohesin phosphorylation could be used to sensitize cancer cells to chemotherapeutics similar to ATM inhibition but ideally with reduced toxicity”. It is not clear what the justification for this sentence is, particularly the comment about reduced toxicity. As such, it should probably be removed. Similarly, the previous sentence about A-T and ATR-X is confusing – while A-T is caused by ATM deficiency, ATR-X is a totally different disease; were the authors thinking that it was caused by ATR deficiency (it is not).

Reviewer #3

(Remarks to the Author)

I have read the manuscript “A Dual Role of Cohesin in DNA DSB Repair” By Michael Fedkenheuer et al with initial interest and curiosity. The authors set out to decipher the roles of Cohesin mediated chromatin loop extrusion and the role as a local DNA repair factor via phosphorylation of the Cohesin subunit Smc1 for DNA double strand break (DSB) repair, and claim that indeed two discrete roles for the separate functions of Cohesin can be understood from their results. After having read the manuscript carefully I am not completely convinced, but this could also be due to the possibly not optimal presentation of the data in the manuscript.

A major concern I have is the procedure and controls for degradation of Rad21 through a mAID degenon system. The cells used are created and published and that is all fine. However I strongly recommend to show the degree of degradation in the hands of the investigators. It is claimed in the method section that this is done routinely by FACS, thus there is no reason to not show the results. Preferably for all experiments where this method is used. Furthermore, although the advantage of the AID system over the siRNA or RNAi based removal of target proteins has been established lately it is still important to make sure that your cells don't show phenotypes based on the addition of Auxin alone. This could be done by the addition of Auxinole in parallel to Auxin. Or Auxin to cells without the AID tag on the target gene. If this is done or not is not clear from the figures, Figure legends or methods. This should be clarified.

A second important concern is that the authors use the AsiSI system in some and induction of DNA damage through γ -radiation in other experiments. These two are not at all comparable. While AsiSI presumably creates clean DSB γ -radiation certainly does not, rather all types of damage including SSBs and crosslinks. In addition, even damage to proteins can occur. The shift from one system to the other should be indicated in the text and motivated for each experiment.

In addition to my major concern there is a number of smaller and bigger issues that I believe should be handled to improve the manuscript. I list them below as they appear in the manuscript with the line/row number as the indicator for each:

7. The address of the first and corresponding author looks strange/repetitive.
- 35-36. Do you want to use numbers in the text in the abstract? Probably better with First and Second. Since I don't think that you have necessarily shown that the two functions are independent you might also consider reformulating the abstract. More on that below.
46. Check your references throughout, at least 4 and 6 are not about DSBs or DSB repair.
- 48-49. formulation; the sentence “cohesin also regulates the topology of chromosomes by extruding through chromatin and forming DNA loops” sounds as if cohesin is extruding through chromatin. I believe it is chromatin that is extruded through cohesin.
53. Regarding Refs 13-16, you could refer to the first papers here instead. Also, you could introduce Cohesin associated proteins such as NIPBL and PDS5 here since you are later talking about them.
64. “arereadily” should be separated and “disperse” should be dispersed, I assume.
67. You cannot really say “cohesin extrusion” maybe cohesin based loop extrusion or sth similar. Worth mentioning already here is that also Smc5/6 is performing loop extrusion and is essential for DSB repair. Could potentially be brought up in the discussion.
71. Segregation between TADs?
76. The statement MRN activates ATM should be referenced.
78. Reference 35 is a yeast paper –better to refer to a human study I think.
- 87-88. Sentence: “To address this question we here study DNA damage in cells where cohesin is depleted by a degenon system and phosphorylation deficient mutants” I would write: study DNA damage and repair ...either by a degenon system....or in cells harbouring a phosphorylation mutant. Mutant in singular.
91. insert time after same
92. replace both with these. If the functions are distinct but complementary have you then really separated them in their importance for DSB repair?
98. spell out ER first time
- 99-100. Show the IF in the absence of 4OHT (or presence of solvent) treatment as negative control.
101. In Fig S1a there is no location shown for the AsiSI breaks. Should also Fig 1b be mentioned? 102. Also difficult to understand the numbers. It says 80-85 AsiSI breaks in the text in Fig S1a it says 1219 total breaks. I understand that these are the potential sites due to sequence - but this could be better explained. Also for Fig 1 (and all) spell out abbreviation NT once.
121. An alternative for investigation of where at the DSB Cohesin loads and ends up, is to utilise an Smc3-ATP hydrolysis mutant. This type of Cohesin mutant was shown to accumulate at DSB ends as opposed to wt cohesin that is found at its normal binding sites (Scherzer et al 2022).

125. I assume you want to say Figure 1f instead of 2b here.
137. replace cohesin extrusion with cohesin based loop extrusion or sth similar
138. Introduce the Rad21- cell line, confirm that it is healthy, that degradation of Rad21 works what is the time required for degradation. Is it complete? And what is your control WT cells plus Auxin or Rad21 AID cells plus Auxinol?
146. increase in inter chromosomal contacts?
157. Motivate why you abandon the specific DSB system for γ -radiation due to the reasons as mentioned above. Also when you use γ -radiation use a unit for radiation that can be understood by most. What does 2.5 mean?
163- Are you here using AsiSI breaks again. Then you are comparing clean DSB with all types of damages. Thus, you have not really tested if H2AX requires loop extrusion – or I need a more clear explanation here.
173. Is the degron system in ES cells the same as used previously in the paper is the degradation of Rad21 comparable. There is no description of the ES cells in the methods as stated.
176. reference 45 – isn't that what you also see and show in fig S1f then maybe indicate that.
187. I am not sure you can call Palbociclib a G1 checkpoint inhibitor. It is a CDK4/6 inhibitor and without CDK4/6 activity the cells will arrest in G1. If you inhibit a checkpoint it rather sounds like the cells would continue through the cell cycle.
224. I recommend to be slightly more humble and modify demonstrate.
230- I am not sure that you have separated the loop extrusion function and the P-repair factor function. I need better and clearer arguments for that. Previous work has before shown that Cohesin protects from repair between distal DSBs (Gelot et al 2016) this should at least be discussed.
234- you say that siRNA systems lead to incomplete knockdown, but as long as you have not shown the degree of degradation of Rad21 - your system is not better. Also, I have not seen a test of the actual loop pattern after Rad21 degradation in your study, making the comparison difficult. You look at intra and inter-chromosomal interactions, but you don't show changes or no changes in loop patterns/positions. In relation to this the paper by Piazza A et al 2021 could be considered to widen the discussion.
289. However cohesinopathy patients don't seem to have an increased risk of Cancer.

Fig 2c. looking t 1h – you might need early events.

Fig 2d. What is the dosage used for damage in the Comet assay? Is the comet assay alkali based or neutral in pH? This decides what type of damage you can say sth about.

Fig 3b. Describe the detection of deletions between breaks better. How big deletions? What is regarded as a deletion? It would also be nice to see a primer pair used that gives 100 % signal when there is no damage induced for comparison.

Fig S2f. Where is the signal for no DSB?

For all of my comments it might be me that have not understood. In such case I apologize. However that means that you need to explain better.

Version 1:

Reviewer comments:

Reviewer #1

(Remarks to the Author)

I thank the Authors for their effort in addressing my concerns.

Given that this manuscript challenges previously published data and models, in my previous report I requested that the authors strengthen the statistical robustness (now adequate) and re-analyze published data. The latter point was only partially addressed, in part due to difficulties in retrieving the original datasets.

Since the Authors' responses cleared most of the issues, I am not providing a point-by-point rebuttal. Yet, I am reporting below some unresolved issues that will require further attention.

Major points

1. The Authors seem to have trouble retrieving some data from Arnould et al., (Nature 2021), particularly the ChIP-seq data for pSMC3 and SCC1. I encourage them to look more carefully in the published repository (E-MTAB-8851): the data should be there since depositing raw and processed data when publishing in Nature Journals is mandatory. Alternatively, I suggest getting in touch with the Legube laboratory for assistance.

2. Concerning the analyses of HiC data, I would ask the Authors to provide (i) averaged Hi-C contact matrix of

$\log_2[+DSB/-DSB]$ centered on the top-induced DSBs (100-200 DSBs, with a 50-kb resolution, 5-Mb window and (ii) the mean aggregate peak analysis (APA) plotted on a 200-kb window (10-kb resolution) before and after DSB induction, calculated between the DSBs and nearby loop anchors, for both wt, RAD21^{-/-}, ATMⁱ and SMC1A-Pdef. These analyses are similar to what was reported by Arnould et al., (Nature 2021); see text figure 2 for reference.

In particular, I am interested in verifying whether these analyses can reveal evidence for loop extrusion at DSBs (as revealed by the “stripes” detected in HiC contact matrices) and then verify whether, if present, loop extrusion depends on cohesins or ATM activity. This is critical evidence to confirm or reject the “loop-extrusion” model.

3. Please summarize in a supplementary figure the evidence that replicates of ChIP-seq data are consistent by displaying the overlap of Peaks detected in replicates and by providing signal-based heatmaps of replicates

Minor points.

1. Line 149. the text reads: “the increase in intra-domain interactions within damaged TADs was minimal—only 1.02-fold compared to control TADs (Figure 2a)”. Is the value reported statistically significant?

2. Line 150. the text reads: “in intra-TAD contacts observed when cohesin was depleted by over 95% in HCT-116 degran cells[43]”. Please confirm that the citation of reference 43 is appropriate.

3. Line 209. The text reads, “This phenotype was easily distinguished by shifts in Cq values when cohesin was depleted (Figure S3c).” Please: (i) correct the typo: distinguished to distinguished and (ii) indicate what Cq is.

4. Line 218. I think it would be more appropriate to refer to Palbociclib as a CDK4,6 inhibitor that arrests cells in G1. Unless I missed some of the experimental details, here Palbociclib inhibits cell cycle progression and not the “G1 checkpoint”.

5. Line 275. “Thus, it appears that cohesin-mediated loops restrict DSBs to specific chromosomal regions,” It is unclear what this means. Are, by any chance, the Authors referring to the repair of the DSBs?

6. Figure 4c: Please indicate what are the values displayed in the color bar

Some comments on the Authors’ rebuttal:

1. Concerning the discrepancy of their data and Arnould et al., (Nature 2021), I do not think that the difference in how cohesins were eliminated (degron vs RNAi) can justify these discrepancies.

2. Concerning the spike-in controls, these are not only useful for a normalization per cell but can also account for differences in ChIP efficiency, which are not always resolved by Library-based normalization.

Reviewer #2

(Remarks to the Author)

The authors addressed the concerns that I raised in my first review.

Reviewer #3

(Remarks to the Author)

The revised manuscript by Fedkenheuer et al, that I have now carefully read, is a much improved version compared to the initial one. Based on the new experiments, clarifications and corrections added, together with the authors answers to my comments, I have no further comments, questions or concerns about the work.

Version 2:

Reviewer comments:

Reviewer #1

(Remarks to the Author)

I thank the authors for making the extra-effort to address my last concerns.

I have to say that I am surprised at the outcome of the average HiC contact matrix analyses (figure 3 of rebuttal), not only the authors are not detecting “stripes” in their HiC datasets (hence there is no evidence of loop extrusion in their experiments) but, at least judging from the figure, it looks to me that they failed to detect stripes also in the re-analysed data from Arnould, et al. (Loop extrusion as a mechanism for formation of DNA damage repair foci. Nature 590, 660–665 (2021). <https://doi.org/10.1038/s41586-021-03193-z>), contrary to what shown in figure 2B of this paper.

It is also perplexing that the re-analyses of ChIP-seq data from Arnould et al. revealed minimal accumulation of cohesin at DSBs.

I have no additional comments.

We sincerely thank the Reviewers for their thorough and insightful evaluation of our work. Addressing their suggestions has significantly enhanced the quality of the manuscript. Below, we provide detailed responses to each of their questions and comments.

Reviewer #1

In this manuscript by Casellas and collaborators, the Authors analyze the role of cohesins in the DNA damage response and in the repair of DNA double strand breaks. They set out to address how cohesins may control chromatin architecture and chromatin loop extrusions upon DNA damage, how cohesins may control the DNA damage response and the spreading of gH2Ax domains intra and inter-TADs, and the role of cohesins in DNA repair and genome instability. They took advantage of the ER-AsISI system to introduce programmed double strand breaks in cell lines engineered with either a degradable RAD21-AID knock-in (for loss of function studies) or a mutated SMC1A (SMC1A-Pdef, lacking ATM dependent phosphorylation site), thus enabling the researchers to potentially sort out cohesins activities controlled by ATM phosphorylation.

The Authors show that Rad21 does not localize at DSBs, yet DSBs stimulate Rad21 binding to damaged regions (TADs), and within these regions Rad21 accumulates more prominently at anchor sites decorated by CTCF. Yet, increased cohesins at damaged TADs has a marginal effect on intra-TAD contacts (only a slight increase is reported). Based on the lack of localization of cohesins at DSBs and lack of a strong increase in intra-TAD contacts the Authors exclude that cohesins may promote loop extrusion at double strand breaks. Also, the spreading of gH2Ax domain was independent of cohesin. They also provide evidence that cohesins are needed to prevent long-distance chromosomal contacts and promote conservative DNA repair (and prevent chromosomal translocations). This latter function depends on Pi of SMCA1.

Importantly, these data are in stark contrast to what was recently proposed by Arnould et. al, (Nature 2021) where cohesins were shown to orchestrate the remodeling of DNA damaged TADs by (i) binding DNA-broken ends present at DSBs, (ii) by promoting loop extrusion, (iii) driving gH2Ax spreading (intra-TAD) and (iii) limiting gH2Ax spreading to adjacent TADs. All these findings and the underlying model are strongly challenged by the work of Casellas, which fails to confirm the main data.

Also, the data in the Casellas' manuscript suggests a role for cohesins in preventing chromosomal translocation, a conclusion that seems to put into question what reported by Arnould and collaborators (Nature 2023), that showed that loss of cohesins reduced non-conservative rejoining of DNA breaks (thus decreasing chromosomal translocations).

These inconsistencies are perplexing and not easily explained considering that the Casellas group used cellular models and experimental strategies that are similar to what was previously employed by Arnould et. al, (Nature 2021 and 2023). If confirmed, the Casellas data would raise serious concerns on the previously published works.

General points

The manuscript needs to be improved in terms of statistical robustness in order to be able

to make a strong case, thus replicates are needed for all NGS analyses. Also, considering that this work strongly challenges previous work, re-analysis of published data is suggested.

Major points:

1. All ChIP-seq need replicates and possibly the use of spike in (as used in Arnaud et. Al, Nature 2021). At present, it appears that analyses were performed on a single sample for each condition (based on the data deposited in GEO by the Authors).

Given the inconsistencies between our findings and previously published results from Legube et al., Reviewer 1 appropriately suggested additional experiments to verify the reproducibility of our data. As a response, we have repeated all ChIP-seq experiments, and the complete dataset is included in the GEO submission.

In our experience, spike-ins are crucial for deep sequencing experiments in two scenarios: (1) when there are broad changes between samples, and (2) when there is no inherent background signal to normalize the samples, enabling side-by-side comparisons. A notable example is the activation of quiescent lymphocytes, where the transcriptome of G0 cells is globally and proportionally amplified as they enter the cell cycle. This amplification can only be accurately visualized using spike-in controls that normalize data on a per-cell basis [1].

In contrast, most other experiments show changes limited to smaller genomic regions. Since most peaks remain unchanged and the IP background (the "lawn" seen between peaks in the genome browser) stays consistent, normalizing based on total reads offers the simplest method for comparing samples. In the few instances where we normalized ChIP-seq experiments with spike-in controls, we found no significant differences in the results. In fact, the accuracy was sometimes compromised due to pipetting errors in the spike-ins, where the small introduced variances were greater than the differences we aimed to detect.

The reviewer also requested a re-analysis of the published data by Legube et al. Regarding cohesin binding, the dataset from Arnould et al., 2021 included only four samples: two inputs (pre-IP controls) and two 4OHT-treated samples precipitated with either pSMC1 or pSMC3 antibodies[2]. Unfortunately, the key control—untreated (no 4OHT) samples—was not provided, making it impossible to normalize the data for a side-by-side comparison with our results. Additionally, the published data covered a limited number of DSBs, specifically those mapped to chromosomes 1 and 6 because this was done using ChIP-chip rather than ChIPseq.

On the other hand, the Hi-C data from the two studies was complete and could be compared[2, 3]. We found no significant discrepancies between our findings and theirs. In line with our main conclusions, the local architectural changes they observed in response to DNA damage, while statistically significant, were very small. The magnitude of these changes should be viewed in the context of experiments where cohesin is depleted, leading to the complete disruption of local architecture. In such cases, only modest transcriptional changes at a few genes have been reported. Therefore, the Hi-C "discrepancies" likely stem from differing interpretations of what may be physiologically relevant.

Two additional discrepancies were noted. First, the claim that γ H2AX foci do not form properly when cohesin is removed contradicts their previous publication using the same system[4], where they reported that the loss of cohesin actually increases γ H2AX signals. Our data, by contrast, showed only negligible differences in the number of γ H2AX foci. Second, the assertion that cohesin removal does not impact the rate of translocations but instead reduces the formation of deletions *in cis* is also contradicted by our findings, as well as those of Gelot et al. in U2OS cells[5].

While the reason for the discrepancy between their new data and both their prior publication and our results remains unclear, we can propose at least two potential explanations:

1. **AsiSI Retroviral Construct Silencing:** One possibility is the gradual silencing of the AsiSI retroviral construct in mammalian cells. In our experience, the AsiSI construct, like most lentiviruses, becomes increasingly silenced over time, likely due to DNA methylation. To mitigate this, we included a P2A-BFP cassette downstream of AsiSI to monitor transgene expression across experimental conditions and cell lines. In some cases, to maintain consistent expression levels, we had to sort the positive cells.
2. **Cohesin Depletion Variability:** Cohesin depletion may also vary between cell lines. While Legube and colleagues used siRNAs to downregulate cohesin, we employed the degron system developed by Kanemaki and colleagues[6]. Both approaches can yield phenotypes resembling a genetic knockout, but achieving this level of depletion is rare and often requires screening numerous siRNAs. Similarly, the effectiveness of the degron system often depends on whether the N- or C-terminus is tagged, and for certain proteins, degradation is always ineffective due to partial occlusion of the termini.

In conclusion, without further insight into their system, it is challenging to explain these contradictions, which is why we have generally refrained from commenting on their results in our manuscript.

2. Authors need to provide evidence for the consistency of DSBs detected in the different cellular models and among replicates. In particular, they need to run replicates of the MER11 ChIPseq in WT cells and SMCA1Pdef cells. Also, since many of the analyses rely on the assumption that DSBs identified on WT cells will also be conserved in Lig4^{-/-} cells, it is also needed to perform the MRE11 ChIPseq in Lig4^{-/-} cells (in duplicates).

Figure S1a and S1c now show the reproducibility of DNA DSBs and ChIP-Seq among different HCT-116 clones expressing AsiSI, including SMCA^{Pdef} and Lig4^{-/-} cells (all done in duplicates but one of the two samples is shown). All these deep-sequencing samples are now included in the GEO submission. As a way of clarification, and as indicated in the Methods section, in all analyses that were based on DSB sites, MRE11 tracks specific to the given genetic background and/or treatment were used.

2. Please provide, as supplemental data (excel file) or as processed data in GEO (bed file), a list with the genomic co-ordinates of ChIPseq peaks called in all the replicates.

The requested excel file is included as supplementary data, listing all ChIP-Seq peaks called in all replicates.

Minor points

1. Fig1.B (and all other figures): when reporting results of the statistical tests, please indicate the groups/conditions evaluated.

We adjusted how groups and conditions were referenced and standardized this throughout figures. Further information can be found in the legend for Figure 1.

2. Fig1b: are the reported Rad21 signals calculated from Rad21 peaks or they are the average Rad21 signal in the TAD?

RAD21 signal was calculated from peaks. This was clarified in the legend for Figure 1 and the results section text.

3. Fig1b,d: please provide a ranked heatmap of the Rad21 signals for all the groups reported in the box plot.

Heatmaps are shown below for all categories. These were not included in the manuscript but we can try to find room in the supplementary if the reviewer thinks they are critical.

Figure R1: Paired heatmaps for boxplots in figure 1b.

Figure R2: Paired heatmaps for boxplots in figure 1d

4. Line 107: the statement is supported by the observation of a single region, thus hardly generalized, can the Authors extend this analysis to all the DSBs?

This statement is now supported by the global analysis of all DSBs in Figure 1d. We have adjusted the text to make this clearer. We thank the reviewer for this clarification.

5. Line 119: Auth. state that accumulation of Rad21 at anchor sites depends on SMC1A phosphorylation, yet it is unclear whether the DSBs found in WT cells are also found in SMC1A Pdef cells. Please provide evidence in support of DSBs consistencies in the different cellular models used.

As shown now in Figure S1a, the results show that breaks are found in both WT and SMC1A^{Pdef} cells.

6. Fig 1F: Rad21 recruitment to damaged DNA is slow compared to what commonly observed with other proteins recruited to DNA breaks. Do the Authors have an explanation? Can they provide some additional positive (a GFP-labelled DDR protein, which are usually recruited fast, or GFP labelled-repair factors, which usually show delayed recruitment) and also a negative control (GFP alone)?

This is an excellent question, one that was also raised by Reviewer 2. The delayed recruitment of cohesin to damaged sites is well-documented in the literature[7], and our results are consistent with these findings. Similarly, the dynamics of H2AX phosphorylation observed in our experiments (see box plot below) align with those reported in other cell types[8]. This consistency supports the conclusion that the DNA repair machinery in HCT116 cells functions as expected and is not atypical.

Figure R3: This panel tracks γ H2AX foci intensity as a marker for a factor that is recruited rapidly to foci. Foci form very quickly and reach peak intensity quickly which is in contrast to the slower accumulation profile of cohesin.

7. Fig 2a and b: while the analysis shows differences of the average of the distributions, this analysis does not show whether, for each gH2Ax domain considered, there is indeed (i) a decrease of intra-domain contacts and (ii) an increase of inter-domain contacts. This is especially relevant for the analysis displayed in fig 2b where, given the high dispersion of the Rad21 $-/-$ data, there could be a relevant fraction of gH2Ax domains in which interdomain interactions are instead decreasing. Thus, the Authors should complement these analyses with evidence of the fraction of gH2Ax regions that show decrease of intra-domain interactions and increase of inter-domain interactions upon Rad21 loss.

To address this question directly, we have included a table below showing the fraction of TADs that exhibit no change, increased, or decreased intradomain and interdomain interactions following DNA damage. As expected, the results align with the overall trends depicted in the box plots in Figures 2a and 2b. Importantly, by including the fold change for these three categories, we can more clearly highlight the point made in response to

question #1—that these changes are generally minimal, except in the case where cohesin was depleted. We appreciate the reviewer’s suggestion for this clarification.

Table R1: Cumulative intra-domain contacts within DSB domains were stratified into 3 categories same (FC = 1 +/- 0.02), increased (FC > 1 + 0.02), or decreased (FC < 1 - 0.02).

Contacts	(%) Intra-domain WTDSB/NT	(%) Intra-domain RAD21-DSB/NT	(%) Intra-domain SMC1A ^{Pdef} DSB/NT	(%) Intra-domain ATMiDSB/NT
Increased	42 (FC = 1.05)	3 (FC = 1.15)	53 (FC = 1.06)	25 (FC = 1.02)
Same	46 (FC = 1.00)	3 (FC = 1.01)	28 (FC = 1.00)	35 (FC = 1.00)
Decreased	12 (FC = 0.97)	94 (FC = 0.78)	19 (FC = 0.96)	40 (FC = 0.97)

Table R2: Cumulative inter-domain contacts between DSB-DSB domains were stratified into 3 categories same (FC = 1 +/- 0.1), increased (FC > 1 + 0.1), or decreased (FC < 1 - 0.1).

Contacts	(%) Inter-domain WTDSB/NT	(%) Inter-domain RAD21-DSB/NT	(%) Inter-domain SMC1A ^{Pdef} DSB/NT	(%) Inter-domain ATMiDSB/NT
Increased	21 (FC = 1.18)	50 (FC = 1.42)	28 (FC = 1.25)	22 (FC = 1.23)
Same	70 (FC = 1.01)	25 (FC = 1.01)	55 (FC = 1.00)	56 (FC = 0.99)
Decreased	9 (FC = 0.78)	25 (FC = 0.79)	17 (FC = 0.79)	22 (FC = 0.84)

Reviewer #2

It has been known for over two decades that SMC1A and SMC3 are substrates of the ATM (and ATR) protein kinase and are phosphorylated on defined serines in response to DNA damage. Mutations of the target sites in SMC1A results in abnormal responses to DNA breakage, including abnormalities in cell cycle checkpoints and enhanced chromosomal breakage after ionizing irradiation. It has remained unclear, however, how this DDR role of SMC1A/SMC3 (cohesin) is related to the known major role of cohesin in regulating higher order chromatin interactions. In this submission, Fedkenheuer et al. perform a series of experiments to try to address this gap in understanding. They provide evidence for a dual role of the cohesin complex in helping cells deal with DNA breakage, one related to ATM/ATR phosphorylation of SMC1A and a second role in insulating damaged DNA/chromatin from long-range interactions, thus influencing the formation of translocations and intra-chromosomal deletions – the latter insight represents the major advance from this body of work. Some clever experimental approaches are used to separate out these functional roles and the insights gained represent a novel addition to the literature. However, the mechanistic insights gained are still somewhat modest.

Specific Comments:

1. Some nice approaches are used to address specific questions, such as the AsiSI system to introduce defined sites of DNA breaks and combining this system with Lig4-deficient cells to prevent re-cutting, which has been a challenging issue with inducible nuclease systems like this. In addition, the application of HiC and HTGTS technologies to these questions represent an important approach and enabled the bottom-line conclusions.

We thank the reviewer for recognizing the value of the techniques employed in the manuscript.

2. Several of the results confirmed insights that were already in the literature. The main novel insight relates to the HTGTS results shown in Fig. 4. Given the importance of the results of this assay and the relatively small quantitative effects of some of their modulations in other assays, they should better quantitate the differences in translocations under the different conditions, including adding the ATMi and SMC1A^{Pdef} curves to Fig. 4D.

We thank the reviewer for this excellent suggestion. We have revised Figure 4d to include contacts between damaged sites in the Pdef and ATMi samples. Because of the number of samples, instead of using a histogram plot as in our previous version of the manuscript, we have opted for a box plot to help visualization. As expected, the RAD21- sample shows a marked increase in contacts, followed by SMC1A^{Pdef} and ATMi. We thank the reviewer for requesting this.

Figure 4d: Box plot showing the density of interchromosomal interactions at translocating sites were calculated for WT, SMC1A^{P^{def}}, ATM inhibited, and RAD21- cells in the presence of AsiSI DSBs. Increased interactions at translocated sites were observed for both phosphorylation deficient and cohesin depleted conditions.

3. They relied to a certain extent on disappearance of gH2AX foci as a surrogate marker for DNA repair. While this is a commonly used approach in the literature, it is somewhat problematic in that it is only a surrogate for repair of DNA breaks and can be misleading; for example, inhibition of gH2AX de-phosphorylation would result in failure of disappearance of the foci but is not representative a failure to repair (examples of this happening are in the literature). To the authors' credit, they acknowledge that foci disappearance is only a surrogate and address this with the comet assay in Fig. 2D/E. Since they have the AsiSI system in place, they have the capability of directly measuring DNA break repair as re-ligation of the broken sites; it would be elegant to also use this approach directly measure DNA break repair if possible.

To further investigate DNA break repair across different genetic backgrounds, we have included an analysis of microhomology (MH) usage in Figure S4d. While this analysis does not measure the number of correct end-joining events, it provides an indication of incorrect repair events. The results, shown below, reveal a significant increase in microhomology-mediated end-joining (MHMEJ) repair in the absence of ligase 4, as previously reported[9]. Interestingly, this effect was mitigated in the absence of cohesin, possibly due to fewer repair events being completed correctly in that genetic background. This result is consistent with one of the main conclusions of the manuscript that cohesin plays a key role in the proper repair of DNA breaks.

Figure S4d: Microhomology usage for non-AsiSI junctions.

4. Many of the effects are quantitatively quite small; they might be statistically significant, but one wonders about the biologic significance of some of these differences. Some comments from the authors on these particular issues would be appreciated. The good news is that that translocation effects are fairly convincing.

We fully agree with the reviewer that some of the effects we observe, particularly regarding the impact of DNA breaks on cohesin extrusion and TAD formation, are quantitatively small, including the one discussed in point #7 below. This is precisely where our interpretation diverges from recent publications. As we discuss in response to question #5 below and in the main text, we do not believe these minor effects are the driving force behind DNA repair. Instead, we propose that ATM phosphorylation imparts the cohesin complex with a new function that, either directly or indirectly, facilitates the repair of DNA lesions.

5. One aspect of their proposed model of how SMC1A/cohesin is involved in DNA breakage responses that remains unclear is whether SMC1A/cohesin is being recruited to the break sites or whether cohesin complexes that are already in place near where breaks occur are affecting long-range interactions. If the former, this appears to not require ATM or SMC1A phosphorylation. Could the authors provide some clarity on their thinking in this regard (both what may be concluded or what remains to be elucidated)?

This is an excellent question that requires several new experiments. As now stated in the discussion, the new results indicate:

- 1- DNA damage triggers cohesin phosphorylation and markedly increases its recruitment around damaged TADs.
- 2- The local increase in cohesin does not lead to a proportional increase in cohesin extrusion (intra-TAD contacts), which only show a minimal increase.
- 3- Cells expressing SMC1A phosphorylation mutants do not exhibit changes in chromatin topology. Instead, they display a marked defect in repair, comparable to cells treated with ATM kinase inhibitors.

- 4- Cells lacking cohesin accumulate approximately twice as many translocations compared to SMC1A phosphorylation mutants, suggesting that cohesin's architectural role extends beyond DNA repair, protecting chromosomes from aberrant joining.

Based on these findings we favor a model in which:

- 1- cohesin is rapidly phosphorylated by ATM in response to DNA breaks.
- 2- Phosphorylated cohesin is likely retained around break sites via protein-protein interactions with phosphorylated repair factors.
- 3- Phosphorylated cohesin aids in DNA lesion repair independently of its function in chromatin extrusion and TAD formation.
- 4- Cohesin's architectural function helps prevent intra-chromosomal deletions and inter-chromosomal translocations by maintaining DNA breaks apart. This activity appears to operate independently of its direct role in the repair of DNA breaks.

Several experiments could further clarify this model. For point #2, generating SMC1A deletion and/or point mutants fused to GFP could help identify which domains are essential for retention at break sites. Once these domains are identified, mass spectrometry could be used to detect cohesin binding partners from irradiated cell extracts. Addressing point #3 is more challenging, as it remains an open question for many repair factors, including ATM. Nevertheless, one could investigate whether cohesin operates upstream or downstream of other factors involved in the repair cascade, similar to studies conducted on other proteins involved in DSB repair.

6. Could the authors comment on how these insights link to the published data showing that SMC1A phosphomutants exhibit significantly increased chromosomal breakage after irradiation, similar that seen in ATM-deficient cells? Are these observations consistent with their data?

Kastan and coworkers have shown that cells expressing SMC1A phosphomutants experience persistent chromosomal aberrations after IR, similar to ATM-deficient cells[10]. Their findings are fully consistent with our results. In our study, we demonstrated through both γ H2AX staining and the comet assay that DNA breaks persist for longer periods in cells lacking cohesin phosphorylation compared to WT cells. We believe this prolonged presence of unrepaired DNA damage underlies the increase in chromosomal translocations observed in these mutants.

7. Data in Fig. 2 suggests that ATM kinase inhibition reduces intra-domain contacts, but that this effect is independent of SMC1A phosphorylation and independent of DNA damage. This observation seems confusing relative to their model and requires some explanation.

The reviewer is correct. According to Figure 2a, there is a 1.02-fold change difference between the undamaged WT sample and the ATMi-treated sample(s). While statistically significant, this change is minor compared to the 0.78-fold difference in intra-TAD

contacts between WT and RAD21- samples (also shown in the same panel). We currently lack a clear explanation for this effect. We have discussed these possibilities in the results. In line with this scenario, SMC1A phosphorylation-deficient samples behave similarly to WT (Figure 2a, main text). Furthermore, multiple cohesin subunits are phosphorylated in mammalian cells[11], but their specific effects on cohesin extrusion remain largely unknown.

8. The last sentence in the discussion states that “.....inhibitors of cohesin phosphorylation could be used to sensitize cancer cells to chemotherapeutics similar to ATM inhibition but ideally with reduced toxicity”. It is not clear what the justification for this sentence is, particularly the comment about reduced toxicity. As such, it should probably be removed. Similarly, the previous sentence about A-T and ATR-X is confusing – while A-T is caused by ATM deficiency, ATR-X is a totally different disease; were the authors thinking that it was caused by ATR deficiency (it is not).

We agree with the reviewer that these sentences were confusing. We have therefore removed them from the manuscript.

Reviewer #3

I have read the manuscript “A Dual Role of Cohesin in DNA DSB Repair” By Michael Fedkenheuer et al with initial interest and curiosity. The authors set out to decipher the roles of Cohesin mediated chromatin loop extrusion and the role as a local DNA repair factor via phosphorylation of the Cohesin subunit Smc1 for DNA double strand break (DSB) repair, and claim that indeed two discrete roles for the separate functions of Cohesin can be understood from their results. After having read the manuscript carefully I am not completely convinced, but this could also be due to the possibly not optimal presentation of the data in the manuscript.

A major concern I have is the procedure and controls for degradation of Rad21 through a mAID degenon system. The cells used are created and published and that is all fine. However I strongly recommend to show the degree of degradation in the hands of the investigators. It is claimed in the method section that this is done routinely by FACS, thus there is no reason to not show the results. Preferably for all experiments where this method is used.

We appreciate Reviewer 3's comments on this point. To provide further clarity, we have included Figure S2a, which illustrates RAD21 depletion in both WT and Ligase 4^{-/-} cells. This figure also features an APA analysis that assesses loops before and after RAD21 depletion, confirming the global loss of architecture. Additionally, we have added immunofluorescence microscopy panels for the mES degenon in Figure S2f.

Figure S2a: RAD21 can be efficiently degraded by the addition of indole 3 acetic acid (IAA) for 6h (top panels) which we measured via loss of RAD21-GFP signal. This degradation leads to inactivation of the cohesin complex resulting in loss of genomic loops visualized by APA analyses (bottom panels).

Figure S2f left panel: The left panel shows an example of RAD21 depletion in mES cells using the same concentration and timing as HCT116. Since RAD21 is fused to a Halo tag, these cells were imaged with the Janelia Fluro 549 ligand.

Furthermore, although the advantage of the AID system over the siRNA or RNAi based removal of target proteins has been established lately it is still important to make sure that your cells don't show phenotypes based on the addition of Auxin alone. This could be done by the addition of Auxinole in parallel to Auxin. Or Auxin to cells without the AID tag on the target gene. If this is done or not is not clear from the figures, Figure legends or methods. This should be clarified.

We appreciate the reviewer for highlighting this issue. Auxin has been used in several DNA damage-related studies at similar concentrations without any reported side effects[12, 13]. However, to strengthen that conclusion we have included additional controls in the new version of the manuscript.

First, we aimed to determine whether auxin or RAD21 depletion induces double-strand breaks (DSBs). This control is included in Figure 2c, showing RAD21 depletion at 0 hours in HCT cells, and in Figure S2f for mES cells. We have also added a no DSB control for RAD21 depletion with auxin in Figure 3b, demonstrating that neither treatment causes AsISI-based deletions.

Second, we wanted to ensure that auxin does not affect DSB repair. While having the auxinole control would be ideal, the concentration required to counteract the 750 μ M auxin (approximately 2.5 mM) induce significant toxicity and cell death. To rule out that auxin affects repair independently of cohesin, we have also included a WT control (no RAD21 degron, no OsTIR1) below.

Figure R4: HCT116 cells without components of the degron system were used as a control for delayed repair assays. These cells were treated with 500 μ M auxin for 6h prior to treatment with 2.5 Gy γ -irradiation. There is no difference in repair between these conditions

A second important concern is that the authors use the AsiSI system in some and induction of DNA damage through γ -radiation in other experiments. These two are not all comparable. While AsiSI presumably creates clean DSB γ -radiation certainly does not, rather all types of damage including SSBs and crosslinks. In addition, even damage to proteins can occur. The shift from one system to the other should be indicated in the text and motivated for each experiment.

We appreciate the reviewer for bringing this oversight to our attention. To address this, we have revised the text in the results section to clarify when certain experiments are not feasible using a continuous DNA damage system. Additionally, we have included more details about the COMET assay methods and specified that we utilized neutral conditions to visualize DNA damage specifically resulting from DSBs.

In addition to my major concern there is a number of smaller and bigger issues that I believe should be handled to improve the manuscript. I list them below as they appear in the manuscript with the line/row number as the indicator for each:

7. The address of the first and corresponding author looks strange/repetitive.

Thank you for catching this error. We have deleted the repetitive text.

35-36. Do you want to use numbers in the text in the abstract? Probably better with First and Second. Since I don't think that you have necessarily shown that the two functions are independent you might also consider reformulating the abstract. More on that below.

-We have changed numbers to text and reworded the abstract. The changes should reflect the message we are trying to convey more clearly.

46. Check your references throughout, at least 4 and 6 are not about DSBs or DSB repair.

We thank the reviewer for pointing this out. these citations have been removed.

48-49. formulation; the sentence “cohesin also regulates the topology of chromosomes by extruding through chromatin and forming DNA loops” sounds as if cohesin is extruding through chromatin. I believe it is chromatin that is extruded through cohesin.

We have clarified the wording in the suggested manner.

53. Regarding Refs 13-16, you could refer to the first papers here instead. Also, you could introduce Cohesin associated proteins such as NIPBL and PDS5 here since you are later talking about them.

We thank the reviewer for this suggestion; however we are unable to revise this section in a manner where citations and ideas are clear. We introduce them instead as two separate points for clarity.

64. “arereadily” should be separated and “disperse” should be dispersed, I assume.

Yes, Fixed.

67. You cannot really say “cohesin extrusion” maybe cohesin based loop extrusion or sth similar.

We have made the suggested change.

Worth mentioning already here is that also Smc5/6 is performing loop extrusion and is essential for DSB repair. Could potentially be brought up in the discussion.

We thank the reviewer for this suggestion and have tried to add this point in, however, it overcomplicates the discussion.

71. Segregation between TADs?

We have clarified the wording.

76. The statement MRN activates ATM should be referenced.

Cited

78. Reference 35 is a yeast paper –better to refer to a human study I think.

Re-cited with a human study

87-88. Sentence: "To address this question we here study DNA damage in cells where cohesin is depleted by a degron system and phosphorylation deficient mutants" I would write: study DNA damage and repair ...either by a degron system....or in cells harbouring a phosphorylation mutant. Mutant in singular.

Agreed, Fixed.

91. insert time after same

Fixed.

92. replace both with these. If the functions are distinct but complementary have you then really separated them in their importance for DSB repair?

Fixed and adjusted language a bit

98. spell out ER first time

Fixed.

99-100. Show the IF in the absence of 4OHT (or presence of solvent) treatment as negative control.

The negative control (4OHT-) is shown on the leftmost panel.

101. In Fig S1a there is no location shown for the AsiSI breaks. Should also Fig 1b be mentioned? 102. Also difficult to understand the numbers. It says 80-85 AsiSI breaks in the text in Fig S1a it says 1219 total breaks. I understand that these are the potential sites due to sequence - but this could be better explained. Also for Fig 1 (and all) spell out abbreviation NT once.

We added supplemental Figure S1a to illustrate the consistency of AsiSI breaks among genotypes in three full chromosome examples. It is also apparent that Ligase 4 and SMC1 (to a lesser degree) have increased MRE11 signal indicating that a higher proportion of the cell population have unrepaired breaks at those sites. Also, NT has been changed to no treatment (NT) in Figure 1b which is the first occurrence.

Figure S1a: MRE11 breaks were consistent among genetic backgrounds as shown in three full chromosome examples. MRE11 signal localized primarily to unmethylated AsiSI sites (red).

121. An alternative for investigation of where at the DSB Cohesin loads and ends up, is to utilise an Smc3-ATP hydrolysis mutant. This type of Cohesin mutant was shown to accumulate at DSB ends as opposed to wt cohesin that is found at its normal binding sites (Scherzer et al 2022).

We appreciate the reviewer's suggestion. We attempted to overexpress extrusion-deficient SMC3 but encountered challenges in achieving an acceptable signal-to-noise ratio. To provide further insight, we have included a more detailed view of the break site in Figure S1c. While we do observe some diffuse cohesin signal around the DSB, it does not appear as distinct peaks.

125. I assume you want to say Figure 1f instead of 2b here.

Yes, we have now corrected that. The text references the correct figure now. Note- Due to changes in the figure layout, this figure is now correctly referenced as figure S2b.

137. replace cohesin extrusion with cohesin based loop extrusion or sth similar

Fixed.

138. Introduce the Rad21- cell line, confirm that it is healthy, that degradation of Rad21 works what is the time required for degradation. Is it complete? And what is your control WT cells plus Auxin or Rad21 AID cells plus Auxinol?

Figure S2a was added to validate cohesin degradation. In Figure 2c, the 0h (non-irradiated) time point for RAD21-, WT, SMC1 is shown. This demonstrates that our conditions and/or treatments do not produce DNA damage. Additionally, we added no damage control to Figure 3b.

146. increase in inter chromosomal contacts?

We clarified this in the text. These are intrachromosomal interactions between all DSBs within a chromosome. This was done for each chromosome individually and compiled. The inter-chromosomal analysis is too computationally expensive to be analyzed in a realistic time.

157. Motivate why you abandon the specific DSB system for γ -radiation due to the reasons as mentioned above. Also when you use γ -radiation use a unit for radiation that can be understood by most. What does 2.5 γ mean?

We thank the reviewer for catching this oversight. We have changed all instance of “2.5” γ to “2.5 Gy γ -irradiation”

163- Are you here using AsiSI breaks again. Then you are comparing clean DSB with all types of damages. Thus, you have not really tested if γ H2AX requires loop extrusion – or I need a more clear explanation here.

We have clarified this further by analyzing foci after AsiSI is induced. We find that γ H2AX foci are present when cohesin is depleted.

Figure R5: The left panel shows foci counts after AsiSI activation with 1 μ M 4OHT for 4h. The right panel shows the GFP profile for these same cells. Cells were fixed, stained and analyzed for γ H2AX foci. In each cell, RAD21-GFP, γ H2AX (A555), and DAPI images were collected and used to produce these plots.

173. Is the degron system in ES cells the same as used previously in the paper is the degradation of Rad21 comparable. There is no description of the ES cells in the methods as stated.

Cohesin depletion in mES is now shown in Figure S2f. We have also clarified the conditions used for depletion in mES cells were the same as those used in HCT116 cells.

176. reference 45 – isn't that what you also see and show in fig S1f then maybe indicate that.

Thank you. We referenced the appropriate figure in addition to the reference 45.

187. I am not sure you can call Palbociclib a G1 checkpoint inhibitor. It is a CDK4/6 inhibitor and without CDK4/6 activity the cells will arrest in G1. If you inhibit a checkpoint it rather sounds like the cells would continue through the cell cycle.

There is ongoing debate regarding the exact mechanism; however, the predominant model suggests that CDK4/6 plays a catalytic role in phosphorylating the retinoblastoma protein (Rb) during G1 phase. Inhibition of CDK4/6 prevents Rb phosphorylation, leading to repression of E2F-mediated transcription. As a result, cells are unable to express the genes necessary for cell cycle entry[14, 15].

We do recognize that this point may require further clarification, so we have included a comparison of EdU incorporation between cycling and non-cycling HCT116 cells in Figure R4.

Figure R6: Edu incorporation for cycling versus non-cycling cells.

224. I recommend to be slightly more humble and modify demonstrate.

We revised this sentence accordingly.

230- I am not sure that you have separated the loop extrusion function and the P-repair factor function. I need better and clearer arguments for that. Previous work has before shown that Cohesin protects from repair between distal DSBs (Gelot et al 2016) this should at least be discussed.

This work is now cited and discussed in the introduction. We thank the reviewer for pointing this out.

234- you say that siRNA systems lead to incomplete knockdown, but as long as you have not shown the degree of degradation of Rad21 - your system is not better. Also, I have not seen a test of the actual loop pattern after Rad21 degradation in your study, making the comparison difficult.

The reviewer is correct; that particular sentence in the original manuscript lacked clarity. We have now revised it to indicate that the discrepancy between our results and those of Legube and colleagues may be attributed to several factors, including the relative effectiveness of RAD21 depletion via siRNA compared to the degron system. In our response to Reviewer 3's initial comments, we have also included histogram plots illustrating the extent of cohesin degradation, as well as APA analyses of contacts between TAD boundaries in the presence and absence of cohesin.

You look at intra and inter-chromosomal interactions, but you don't show changes or no changes in loop patterns/positions. In relation to this the paper by Piazza A et al 2021 could be considered to widen the discussion.

We appreciate the reviewer's excellent suggestions. We have added an analysis of gained and lost loops in the results section. Additionally, the paper by Piazza et al. (2021) supports one of our primary hypotheses, which we discuss at the end of the discussion section and have appropriately cited.

289. However cohesinopathy patients don't seem to have an increased risk of Cancer.

That is correct, we have rectified the sentence in question.

Fig 2c. looking t 1h – you might need early events.

As shown in the modified figure (reproduced below), we have found no major difference in foci at early time points.

Figure R7: A short time point experiment was done at 0, 15, 30, 45, and 60 minutes post treatment of HCT116 cells with 2.5 Gy γ -irradiation. The top two panels show foci intensity in WT and RAD21- cells. The bottom two panels show the foci counts. The final plot shows the cumulative GFP intensity summed between time points to confirm RAD21 depletion.

Fig 2d. What is the dosage used for damage in the Comet assay? Is the comet assay alkali based or neutral in pH? This decides what type of damage you can say sth about.

We thank the reviewer for catching this. We forgot to add the methods for the COMET assay. We further adjusted the text in the results and figure legends to specify the conditions clearly. The assay is done under neutral conditions.

Fig 3b. Describe the detection of deletions between breaks better. How big deletions?

What is regarded as a deletion? It would also be nice to see a primer pair used that gives 100 % signal when there is no damage induced for comparison.

We have included Figures S3a-c to demonstrate the specificity of the qPCR assay. The deletion junctions are already ordered by size, and we have also provided the exact deletion sizes in the methods section, as detailed in Table 1.

Fig S2f. Where is the signal for no DSB?

We have included KAP1 phosphorylation data for mES cells at 5h post γ -irradiation, including the proper controls.

For all of my comments it might be me that have not understood. In such case I apologize. However that means that you need to explain better.

We believe that all reviewers' questions and requests have improved the clarity of the manuscript.

References

1. Kouzine, F., et al., *Global regulation of promoter melting in naive lymphocytes*. Cell, 2013. **153**(5): p. 988-99.
2. Arnould, C., et al., *Loop extrusion as a mechanism for formation of DNA damage repair foci*. Nature, 2021. **590**(7847): p. 660-665.
3. Arnould, C., et al., *Chromatin compartmentalization regulates the response to DNA damage*. Nature, 2023. **623**(7985): p. 183-192.
4. Caron, P., et al., *Cohesin protects genes against gammaH2AX Induced by DNA double-strand breaks*. PLoS Genet, 2012. **8**(1): p. e1002460.
5. Gelot, C., et al., *The Cohesin Complex Prevents the End Joining of Distant DNA Double-Strand Ends*. Mol Cell, 2016. **61**(1): p. 15-26.
6. Natsume, T., et al., *Rapid Protein Depletion in Human Cells by Auxin-Inducible Degron Tagging with Short Homology Donors*. Cell Rep, 2016. **15**(1): p. 210-218.
7. Kong, X., et al., *Distinct functions of human cohesin-SA1 and cohesin-SA2 in double-strand break repair*. Mol Cell Biol, 2014. **34**(4): p. 685-98.
8. Kinner, A., et al., *Gamma-H2AX in recognition and signaling of DNA double-strand breaks in the context of chromatin*. Nucleic Acids Res, 2008. **36**(17): p. 5678-94.
9. Sharma, S., et al., *Homology and enzymatic requirements of microhomology-dependent alternative end joining*. Cell Death Dis, 2015. **6**(3): p. e1697.
10. Kitagawa, R., et al., *Phosphorylation of SMC1 is a critical downstream event in the ATM-NBS1-BRCA1 pathway*. Genes Dev, 2004. **18**(12): p. 1423-38.
11. Bass, T.E., et al., *ATM Regulation of the Cohesin Complex Is Required for Repression of DNA Replication and Transcription in the Vicinity of DNA Double-Strand Breaks*. Mol Cancer Res, 2023. **21**(3): p. 261-273.
12. Mirman, Z., et al., *53BP1-shieldin-dependent DSB processing in BRCA1-deficient cells requires CST-Polalpha-primase fill-in synthesis*. Nat Cell Biol, 2022. **24**(1): p. 51-61.
13. Wu, J., et al., *Cohesin maintains replication timing to suppress DNA damage on cancer genes*. Nat Genet, 2023. **55**(8): p. 1347-1358.
14. Pennycook, B.R. and A.R. Barr, *Palbociclib-mediated cell cycle arrest can occur in the absence of the CDK inhibitors p21 and p27*. Open Biol, 2021. **11**(11): p. 210125.
15. Crozier, L., et al., *CDK4/6 inhibitor-mediated cell overgrowth triggers osmotic and replication stress to promote senescence*. Mol Cell, 2023. **83**(22): p. 4062-4077 e5.

Reviewer #1 (Remarks to the Author):

I thank the Authors for their effort in addressing my concerns.

Given that this manuscript challenges previously published data and models, in my previous report I requested that the authors strengthen the statistical robustness (now adequate) and re-analyze published data. The latter point was only partially addressed, in part due to difficulties in retrieving the original datasets.

Since the Authors' responses cleared most of the issues, I am not providing a point-by-point rebuttal. Yet, I am reporting below some unresolved issues that will require further attention.

Major points

1. The Authors seem to have trouble retrieving some data from Arnould et al., (Nature 2021), particularly the ChIP-seq data for pSMC3 and SCC1. I encourage them to look more carefully in the published repository (E-MTAB-8851): the data should be there since depositing raw and processed data when publishing in Nature Journals is mandatory. Alternatively, I suggest getting in touch with the Legube laboratory for assistance.

We thank the reviewer for bringing this to our attention. We have now located the data in the referenced repository and performed a side-by-side comparison with our data. We find that Legube et al. results are very consistent with ours in that they show a slight increase in cohesin signals around the breakpoints. As shown in Figure 1, cohesin peaks at damaged sites are shallow and diffuse compared to those at CTCF anchors nearby, which are well defined. A composite analysis supports this conclusion showing minimal cohesin binding at DSBs (Figure 2). This pattern could reflect extrusion complexes arresting at damaged sites due to end processing by repair proteins. At the same time, the results are consistent with small increase in intra-TAD contacts upon DNA damage that we report.

Figure 1: Four examples of AsiSI-damaged loci (defined by Mre11 signals). RAD21 peaks at U2OS (Legube et al.) or HCT116 (WT, Lig4^{-/-} our data) cells are shown in the absence (NT) or presence of DSBs (damaged).

Figure 2: ChIPseq analysis of RAD21 intensity at DSB break sites versus the average intensity of CTCF positive cohesin peaks (anchor site).

2. Concerning the analyses of HiC data, I would ask the Authors to provide (i) averaged Hi-C contact matrix of $\log_2[+DSB/-DSB]$ centered on the top-induced DSBs (100-200 DSBs, with a 50-kb resolution, 5-Mb window and (ii) the mean aggregate peak analysis (APA) plotted on a 200-kb window (10-kb resolution) before and after DSB induction, calculated between the DSBs and nearby loop anchors, for both wt, RAD21^{-/-}, ATMⁱ and SMC1A-Pdef. These analyses are similar to what was reported by Arnould et al., (Nature 2021); see text figure 2 for reference.

In particular, I am interested in verifying whether these analyses can reveal evidence for loop extrusion at DSBs (as revealed by the “stripes” detected in HiC contact matrices) and then verify whether, if present, loop extrusion depends on cohesins or ATM activity. This is critical evidence to confirm or reject the “loop-extrusion” model.

Figure 3 reproduces the weak stripes reported by Legube and coworkers. We do not detect them in our HiC maps, perhaps a reflection of the different cell types used or the amount of damage inflicted in the two systems. Stripes at damaged sites could form if cohesin rings extrude chromatin asymmetrically, arrested at the break point and reeling in upstream or downstream sequences[1]. This scenario would be consistent with the small increase in cohesin signals at damaged sites. The APA analysis in Figure 4 confirms the minimal change in contact frequency at damaged TADs. Therefore, both the previously published and the new data show that the architectural impact of DNA damage is modest. However, the role of cohesin in DNA repair per se is substantial. This is why we favor the model where cohesin phosphorylation by ATM regulates repair in a manner that goes beyond chromatin topology.

Figure 3: Averaged Hi-C contact matrix of $\log_2[+DSB/-DSB]$ centered on the top-induced DSBs (100-200 DSBs, with a 50-kb resolution, 5-Mb window).

Figure 4: mean aggregate peak analysis (APA) plotted on a 200-kb window (10-kb resolution) before and after DSB induction, calculated between the DSBs and nearby loop anchors, for WT, RAD21-, ATMi, and SMC1A-Pdef

3. Please summarize in a supplementary figure the evidence that replicates of ChIP-seq data are consistent by displaying the overlap of Peaks detected in replicates and by providing signal-based heatmaps of replicates

As requested, we have displayed the complete analysis of ChIPseq experiments that comprised replicate 2. We find that the FC measured for all experiments adhere to the patterns described in replicate 1 (Figure 5). We show that data clusters first between backgrounds and next between conditions in the MDS plots of cohesin peaks (Figure 6). We have also provided heatmaps for further comparison in Figure 7. We do note differences in intensity between replicates due to sequencing depth; however, FC comparison between damaged and undamaged sites were consistent between replicates of all conditions and backgrounds (Figure 5).

Figure 5: Replicate ChIPseq experiments analyzed in the same way presented in the manuscript. Panel a is consistent with our findings seen in manuscript figure 1b. Panel b is consistent with the findings shown in manuscript figure 1d.

Figure 6: Principal component analysis (PCA) plot calculated using all cohesin peaks. Data for both replicates is shown (replicate 1 circles and replicate 2 triangles)

Figure 7: Heatmaps comprising all RAD21 peaks for treatment conditions and backgrounds for experiments comprising replicates 1 and 2.

Minor points.

1. Line 149. the text reads: " the increase in intra-domain interactions within damaged TADs was minimal—only 1.02-fold compared to control TADs (Figure 2a)". Is the value reported statistically significant?

Yes, the result is significant (DSB/NT P value = $9.2e-05$). This is indicated in the main text now.

2. Line 150. the text reads: "in intra-TAD contacts observed when cohesin was depleted by over 95% in HCT-116 degron cells[43]". Please confirm that the citation of reference 43 is appropriate.

Yes, the citation refers to the efficiency of the RAD21 degradation, but since we had already made this point earlier in the manuscript, we have removed it from this sentence.

3. Line 209. The text reads, "This phenotype was easily distinguished by shifts in Cq values when cohesin was depleted (Figure S3c)." Please: (i) correct the typo: distinguished to distinguished and (ii) indicate what Cq is.

Corrected

4. Line 218. I think it would be more appropriate to refer to Palbociclib as a CDK4,6 inhibitor that arrests cells in G1. Unless I missed some of the experimental details, here Palbociclib inhibits cell cycle progression and not the "G1 checkpoint".

We thank the reviewer for pointing out this confusing statement. This has been corrected.

5. Line 275. "Thus, it appears that cohesin-mediated loops restrict DSBs to specific chromosomal regions," It is unclear what this means. Are, by any chance, the Authors referring to the repair of the DSBs?

We have clarified the sentence with the reviewer's suggestion.

6. Figure 4c: Please indicate what are the values displayed in the color bar

We have added the appropriate clarification to the figure.

Some comments on the Authors' rebuttal:

1. Concerning the discrepancy of their data and Arnould et al., (Nature 2021), I do not think that the difference in how cohesins were eliminated (degron vs RNAi) can justify these discrepancies.

We agree with the reviewer that other options could explain the discrepancies between our results and those of Arnould et al, particularly with respect to the differences in translocations.

2. Concerning the spike-in controls, these are not only useful for a normalization per cell but can also account for differences in ChIP efficiency, which are not always resolved by Library-based normalization.

We thank the reviewer for pointing this out.

References

1. Vian, L., et al., *The Energetics and Physiological Impact of Cohesin Extrusion*. *Cell*, 2018. **173**(5): p. 1165-1178 e20.

We sincerely thank the Reviewers for their thorough and insightful evaluation of our work. Addressing their suggestions has significantly enhanced the quality of the manuscript. Below, we provide detailed responses to each of their questions and comments.

Reviewer #1

In this manuscript by Casellas and collaborators, the Authors analyze the role of cohesins in the DNA damage response and in the repair of DNA double strand breaks. They set out to address how cohesins may control chromatin architecture and chromatin loop extrusions upon DNA damage, how cohesins may control the DNA damage response and the spreading of gH2Ax domains intra and inter-TADs, and the role of cohesins in DNA repair and genome instability. They took advantage of the ER-AsISI system to introduce programmed double strand breaks in cell lines engineered with either a degradable RAD21-AID knock-in (for loss of function studies) or a mutated SMC1A (SMC1A-Pdef, lacking ATM dependent phosphorylation site), thus enabling the researchers to potentially sort out cohesins activities controlled by ATM phosphorylation.

The Authors show that Rad21 does not localize at DSBs, yet DSBs stimulate Rad21 binding to damaged regions (TADs), and within these regions Rad21 accumulates more prominently at anchor sites decorated by CTCF. Yet, increased cohesins at damaged TADs has a marginal effect on intra-TAD contacts (only a slight increase is reported). Based on the lack of localization of cohesins at DSBs and lack of a strong increase in intra-TAD contacts the Authors exclude that cohesins may promote loop extrusion at double strand breaks. Also, the spreading of gH2Ax domain was independent of cohesin. They also provide evidence that cohesins are needed to prevent long-distance chromosomal contacts and promote conservative DNA repair (and prevent chromosomal translocations). This latter function depends on Pi of SMCA1.

Importantly, these data are in stark contrast to what was recently proposed by Arnould et. al, (Nature 2021) where cohesins were shown to orchestrate the remodeling of DNA damaged TADs by (i) binding DNA-broken ends present at DSBs, (ii) by promoting loop extrusion, (iii) driving gH2Ax spreading (intra-TAD) and (iii) limiting gH2Ax spreading to adjacent TADs. All these findings and the underlying model are strongly challenged by the work of Casellas, which fails to confirm the main data.

Also, the data in the Casellas' manuscript suggests a role for cohesins in preventing chromosomal translocation, a conclusion that seems to put into question what reported by Arnould and collaborators (Nature 2023), that showed that loss of cohesins reduced non-conservative rejoining of DNA breaks (thus decreasing chromosomal translocations).

These inconsistencies are perplexing and not easily explained considering that the Casellas group used cellular models and experimental strategies that are similar to what was previously employed by Arnould et. al, (Nature 2021 and 2023). If confirmed, the Casellas data would raise serious concerns on the previously published works.

General points

The manuscript needs to be improved in terms of statistical robustness in order to be able

to make a strong case, thus replicates are needed for all NGS analyses. Also, considering that this work strongly challenges previous work, re-analysis of published data is suggested.

Major points:

1. All ChIP-seq need replicates and possibly the use of spike in (as used in Arnaud et. Al, Nature 2021). At present, it appears that analyses were performed on a single sample for each condition (based on the data deposited in GEO by the Authors).

Given the inconsistencies between our findings and previously published results from Legube et al., Reviewer 1 appropriately suggested additional experiments to verify the reproducibility of our data. As a response, we have repeated all ChIP-seq experiments, and the complete dataset is included in the GEO submission.

In our experience, spike-ins are crucial for deep sequencing experiments in two scenarios: (1) when there are broad changes between samples, and (2) when there is no inherent background signal to normalize the samples, enabling side-by-side comparisons. A notable example is the activation of quiescent lymphocytes, where the transcriptome of G0 cells is globally and proportionally amplified as they enter the cell cycle. This amplification can only be accurately visualized using spike-in controls that normalize data on a per-cell basis [1].

In contrast, most other experiments show changes limited to smaller genomic regions. Since most peaks remain unchanged and the IP background (the "lawn" seen between peaks in the genome browser) stays consistent, normalizing based on total reads offers the simplest method for comparing samples. In the few instances where we normalized ChIP-seq experiments with spike-in controls, we found no significant differences in the results. In fact, the accuracy was sometimes compromised due to pipetting errors in the spike-ins, where the small introduced variances were greater than the differences we aimed to detect.

The reviewer also requested a re-analysis of the published data by Legube et al. Regarding cohesin binding, the dataset from Arnould et al., 2021 included only four samples: two inputs (pre-IP controls) and two 4OHT-treated samples precipitated with either pSMC1 or pSMC3 antibodies[2]. Unfortunately, the key control—untreated (no 4OHT) samples—was not provided, making it impossible to normalize the data for a side-by-side comparison with our results. Additionally, the published data covered a limited number of DSBs, specifically those mapped to chromosomes 1 and 6 because this was done using ChIP-chip rather than ChIPseq.

On the other hand, the Hi-C data from the two studies was complete and could be compared[2, 3]. We found no significant discrepancies between our findings and theirs. In line with our main conclusions, the local architectural changes they observed in response to DNA damage, while statistically significant, were very small. The magnitude of these changes should be viewed in the context of experiments where cohesin is depleted, leading to the complete disruption of local architecture. In such cases, only modest transcriptional changes at a few genes have been reported. Therefore, the Hi-C "discrepancies" likely stem from differing interpretations of what may be physiologically relevant.

Two additional discrepancies were noted. First, the claim that γ H2AX foci do not form properly when cohesin is removed contradicts their previous publication using the same system[4], where they reported that the loss of cohesin actually increases γ H2AX signals. Our data, by contrast, showed only negligible differences in the number of γ H2AX foci. Second, the assertion that cohesin removal does not impact the rate of translocations but instead reduces the formation of deletions *in cis* is also contradicted by our findings, as well as those of Gelot et al. in U2OS cells[5].

While the reason for the discrepancy between their new data and both their prior publication and our results remains unclear, we can propose at least two potential explanations:

1. **AsiSI Retroviral Construct Silencing:** One possibility is the gradual silencing of the AsiSI retroviral construct in mammalian cells. In our experience, the AsiSI construct, like most lentiviruses, becomes increasingly silenced over time, likely due to DNA methylation. To mitigate this, we included a P2A-BFP cassette downstream of AsiSI to monitor transgene expression across experimental conditions and cell lines. In some cases, to maintain consistent expression levels, we had to sort the positive cells.
2. **Cohesin Depletion Variability:** Cohesin depletion may also vary between cell lines. While Legube and colleagues used siRNAs to downregulate cohesin, we employed the degron system developed by Kanemaki and colleagues[6]. Both approaches can yield phenotypes resembling a genetic knockout, but achieving this level of depletion is rare and often requires screening numerous siRNAs. Similarly, the effectiveness of the degron system often depends on whether the N- or C-terminus is tagged, and for certain proteins, degradation is always ineffective due to partial occlusion of the termini.

In conclusion, without further insight into their system, it is challenging to explain these contradictions, which is why we have generally refrained from commenting on their results in our manuscript.

2. Authors need to provide evidence for the consistency of DSBs detected in the different cellular models and among replicates. In particular, they need to run replicates of the MER11 ChIPseq in WT cells and SMCA1Pdef cells. Also, since many of the analyses rely on the assumption that DSBs identified on WT cells will also be conserved in Lig4^{-/-} cells, it is also needed to perform the MRE11 ChIPseq in Lig4^{-/-} cells (in duplicates).

Figure S1a and S1c now show the reproducibility of DNA DSBs and ChIP-Seq among different HCT-116 clones expressing AsiSI, including SMCA^{Pdef} and Lig4^{-/-} cells (all done in duplicates but one of the two samples is shown). All these deep-sequencing samples are now included in the GEO submission. As a way of clarification, and as indicated in the Methods section, in all analyses that were based on DSB sites, MRE11 tracks specific to the given genetic background and/or treatment were used.

2. Please provide, as supplemental data (excel file) or as processed data in GEO (bed file), a list with the genomic co-ordinates of ChIPseq peaks called in all the replicates.

The requested excel file is included as supplementary data, listing all ChIP-Seq peaks called in all replicates.

Minor points

1. Fig1.B (and all other figures): when reporting results of the statistical tests, please indicate the groups/conditions evaluated.

We adjusted how groups and conditions were referenced and standardized this throughout figures. Further information can be found in the legend for Figure 1.

2. Fig1b: are the reported Rad21 signals calculated from Rad21 peaks or they are the average Rad21 signal in the TAD?

RAD21 signal was calculated from peaks. This was clarified in the legend for Figure 1 and the results section text.

3. Fig1b,d: please provide a ranked heatmap of the Rad21 signals for all the groups reported in the box plot.

Heatmaps are shown below for all categories. These were not included in the manuscript but we can try to find room in the supplementary if the reviewer thinks they are critical.

Figure R1: Paired heatmaps for boxplots in figure 1b.

Figure R2: Paired heatmaps for boxplots in figure 1d

4. Line 107: the statement is supported by the observation of a single region, thus hardly generalized, can the Authors extend this analysis to all the DSBs?

This statement is now supported by the global analysis of all DSBs in Figure 1d. We have adjusted the text to make this clearer. We thank the reviewer for this clarification.

5. Line 119: Auth. state that accumulation of Rad21 at anchor sites depends on SMC1A phosphorylation, yet it is unclear whether the DSBs found in WT cells are also found in SMC1A Pdef cells. Please provide evidence in support of DSBs consistencies in the different cellular models used.

As shown now in Figure S1a, the results show that breaks are found in both WT and SMC1A^{Pdef} cells.

6. Fig 1F: Rad21 recruitment to damaged DNA is slow compared to what commonly observed with other proteins recruited to DNA breaks. Do the Authors have an explanation? Can they provide some additional positive (a GFP-labelled DDR protein, which are usually recruited fast, or GFP labelled-repair factors, which usually show delayed recruitment) and also a negative control (GFP alone)?

This is an excellent question, one that was also raised by Reviewer 2. The delayed recruitment of cohesin to damaged sites is well-documented in the literature[7], and our results are consistent with these findings. Similarly, the dynamics of H2AX phosphorylation observed in our experiments (see box plot below) align with those reported in other cell types[8]. This consistency supports the conclusion that the DNA repair machinery in HCT116 cells functions as expected and is not atypical.

Figure R3: This panel tracks γ H2AX foci intensity as a marker for a factor that is recruited rapidly to foci. Foci form very quickly and reach peak intensity quickly which is in contrast to the slower accumulation profile of cohesin.

7. Fig 2a and b: while the analysis shows differences of the average of the distributions, this analysis does not show whether, for each gH2Ax domain considered, there is indeed (i) a decrease of intra-domain contacts and (ii) an increase of inter-domain contacts. This is especially relevant for the analysis displayed in fig 2b where, given the high dispersion of the Rad21 $-/-$ data, there could be a relevant fraction of gH2Ax domains in which interdomain interactions are instead decreasing. Thus, the Authors should complement these analyses with evidence of the fraction of gH2Ax regions that show decrease of intra-domain interactions and increase of inter-domain interactions upon Rad21 loss.

To address this question directly, we have included a table below showing the fraction of TADs that exhibit no change, increased, or decreased intradomain and interdomain interactions following DNA damage. As expected, the results align with the overall trends depicted in the box plots in Figures 2a and 2b. Importantly, by including the fold change for these three categories, we can more clearly highlight the point made in response to

question #1—that these changes are generally minimal, except in the case where cohesin was depleted. We appreciate the reviewer’s suggestion for this clarification.

Table R1: Cumulative intra-domain contacts within DSB domains were stratified into 3 categories same (FC = 1 +/- 0.02), increased (FC > 1 + 0.02), or decreased (FC < 1 - 0.02).

Contacts	(%) Intra-domain WTDSB/NT	(%) Intra-domain RAD21-DSB/NT	(%) Intra-domain SMC1A ^{Pdef} DSB/NT	(%) Intra-domain ATMiDSB/NT
Increased	42 (FC = 1.05)	3 (FC = 1.15)	53 (FC = 1.06)	25 (FC = 1.02)
Same	46 (FC = 1.00)	3 (FC = 1.01)	28 (FC = 1.00)	35 (FC = 1.00)
Decreased	12 (FC = 0.97)	94 (FC = 0.78)	19 (FC = 0.96)	40 (FC = 0.97)

Table R2: Cumulative inter-domain contacts between DSB-DSB domains were stratified into 3 categories same (FC = 1 +/- 0.1), increased (FC > 1 + 0.1), or decreased (FC < 1 - 0.1).

Contacts	(%) Inter-domain WTDSB/NT	(%) Inter-domain RAD21-DSB/NT	(%) Inter-domain SMC1A ^{Pdef} DSB/NT	(%) Inter-domain ATMiDSB/NT
Increased	21 (FC = 1.18)	50 (FC = 1.42)	28 (FC = 1.25)	22 (FC = 1.23)
Same	70 (FC = 1.01)	25 (FC = 1.01)	55 (FC = 1.00)	56 (FC = 0.99)
Decreased	9 (FC = 0.78)	25 (FC = 0.79)	17 (FC = 0.79)	22 (FC = 0.84)

Reviewer #2

It has been known for over two decades that SMC1A and SMC3 are substrates of the ATM (and ATR) protein kinase and are phosphorylated on defined serines in response to DNA damage. Mutations of the target sites in SMC1A results in abnormal responses to DNA breakage, including abnormalities in cell cycle checkpoints and enhanced chromosomal breakage after ionizing irradiation. It has remained unclear, however, how this DDR role of SMC1A/SMC3 (cohesin) is related to the known major role of cohesin in regulating higher order chromatin interactions. In this submission, Fedkenheuer et al. perform a series of experiments to try to address this gap in understanding. They provide evidence for a dual role of the cohesin complex in helping cells deal with DNA breakage, one related to ATM/ATR phosphorylation of SMC1A and a second role in insulating damaged DNA/chromatin from long-range interactions, thus influencing the formation of translocations and intra-chromosomal deletions – the latter insight represents the major advance from this body of work. Some clever experimental approaches are used to separate out these functional roles and the insights gained represent a novel addition to the literature. However, the mechanistic insights gained are still somewhat modest.

Specific Comments:

1. Some nice approaches are used to address specific questions, such as the AsiSI system to introduce defined sites of DNA breaks and combining this system with Lig4-deficient cells to prevent re-cutting, which has been a challenging issue with inducible nuclease systems like this. In addition, the application of HiC and HTGTS technologies to these questions represent an important approach and enabled the bottom-line conclusions.

We thank the reviewer for recognizing the value of the techniques employed in the manuscript.

2. Several of the results confirmed insights that were already in the literature. The main novel insight relates to the HTGTS results shown in Fig. 4. Given the importance of the results of this assay and the relatively small quantitative effects of some of their modulations in other assays, they should better quantitate the differences in translocations under the different conditions, including adding the ATMi and SMC1A^{Pdef} curves to Fig. 4D.

We thank the reviewer for this excellent suggestion. We have revised Figure 4d to include contacts between damaged sites in the Pdef and ATMi samples. Because of the number of samples, instead of using a histogram plot as in our previous version of the manuscript, we have opted for a box plot to help visualization. As expected, the RAD21- sample shows a marked increase in contacts, followed by SMC1A^{Pdef} and ATMi. We thank the reviewer for requesting this.

Figure 4d: Box plot showing the density of interchromosomal interactions at translocating sites were calculated for WT, SMC1A^{Pdef}, ATM inhibited, and RAD21- cells in the presence of AsiSI DSBs. Increased interactions at translocated sites were observed for both phosphorylation deficient and cohesin depleted conditions.

3. They relied to a certain extent on disappearance of gH2AX foci as a surrogate marker for DNA repair. While this is a commonly used approach in the literature, it is somewhat problematic in that it is only a surrogate for repair of DNA breaks and can be misleading; for example, inhibition of gH2AX de-phosphorylation would result in failure of disappearance of the foci but is not representative a failure to repair (examples of this happening are in the literature). To the authors' credit, they acknowledge that foci disappearance is only a surrogate and address this with the comet assay in Fig. 2D/E. Since they have the AsiSI system in place, they have the capability of directly measuring DNA break repair as re-ligation of the broken sites; it would be elegant to also use this approach directly measure DNA break repair if possible.

To further investigate DNA break repair across different genetic backgrounds, we have included an analysis of microhomology (MH) usage in Figure S4d. While this analysis does not measure the number of correct end-joining events, it provides an indication of incorrect repair events. The results, shown below, reveal a significant increase in microhomology-mediated end-joining (MHMEJ) repair in the absence of ligase 4, as previously reported[9]. Interestingly, this effect was mitigated in the absence of cohesin, possibly due to fewer repair events being completed correctly in that genetic background. This result is consistent with one of the main conclusions of the manuscript that cohesin plays a key role in the proper repair of DNA breaks.

Figure S4d: Microhomology usage for non-AsiSI junctions.

4. Many of the effects are quantitatively quite small; they might be statistically significant, but one wonders about the biologic significance of some of these differences. Some comments from the authors on these particular issues would be appreciated. The good news is that that translocation effects are fairly convincing.

We fully agree with the reviewer that some of the effects we observe, particularly regarding the impact of DNA breaks on cohesin extrusion and TAD formation, are quantitatively small, including the one discussed in point #7 below. This is precisely where our interpretation diverges from recent publications. As we discuss in response to question #5 below and in the main text, we do not believe these minor effects are the driving force behind DNA repair. Instead, we propose that ATM phosphorylation imparts the cohesin complex with a new function that, either directly or indirectly, facilitates the repair of DNA lesions.

5. One aspect of their proposed model of how SMC1A/cohesin is involved in DNA breakage responses that remains unclear is whether SMC1A/cohesin is being recruited to the break sites or whether cohesin complexes that are already in place near where breaks occur are affecting long-range interactions. If the former, this appears to not require ATM or SMC1A phosphorylation. Could the authors provide some clarity on their thinking in this regard (both what may be concluded or what remains to be elucidated)?

This is an excellent question that requires several new experiments. As now stated in the discussion, the new results indicate:

- 1- DNA damage triggers cohesin phosphorylation and markedly increases its recruitment around damaged TADs.
- 2- The local increase in cohesin does not lead to a proportional increase in cohesin extrusion (intra-TAD contacts), which only show a minimal increase.
- 3- Cells expressing SMC1A phosphorylation mutants do not exhibit changes in chromatin topology. Instead, they display a marked defect in repair, comparable to cells treated with ATM kinase inhibitors.

- 4- Cells lacking cohesin accumulate approximately twice as many translocations compared to SMC1A phosphorylation mutants, suggesting that cohesin's architectural role extends beyond DNA repair, protecting chromosomes from aberrant joining.

Based on these findings we favor a model in which:

- 1- cohesin is rapidly phosphorylated by ATM in response to DNA breaks.
- 2- Phosphorylated cohesin is likely retained around break sites via protein-protein interactions with phosphorylated repair factors.
- 3- Phosphorylated cohesin aids in DNA lesion repair independently of its function in chromatin extrusion and TAD formation.
- 4- Cohesin's architectural function helps prevent intra-chromosomal deletions and inter-chromosomal translocations by maintaining DNA breaks apart. This activity appears to operate independently of its direct role in the repair of DNA breaks.

Several experiments could further clarify this model. For point #2, generating SMC1A deletion and/or point mutants fused to GFP could help identify which domains are essential for retention at break sites. Once these domains are identified, mass spectrometry could be used to detect cohesin binding partners from irradiated cell extracts. Addressing point #3 is more challenging, as it remains an open question for many repair factors, including ATM. Nevertheless, one could investigate whether cohesin operates upstream or downstream of other factors involved in the repair cascade, similar to studies conducted on other proteins involved in DSB repair.

6. Could the authors comment on how these insights link to the published data showing that SMC1A phosphomutants exhibit significantly increased chromosomal breakage after irradiation, similar that seen in ATM-deficient cells? Are these observations consistent with their data?

Kastan and coworkers have shown that cells expressing SMC1A phosphomutants experience persistent chromosomal aberrations after IR, similar to ATM-deficient cells[10]. Their findings are fully consistent with our results. In our study, we demonstrated through both γ H2AX staining and the comet assay that DNA breaks persist for longer periods in cells lacking cohesin phosphorylation compared to WT cells. We believe this prolonged presence of unrepaired DNA damage underlies the increase in chromosomal translocations observed in these mutants.

7. Data in Fig. 2 suggests that ATM kinase inhibition reduces intra-domain contacts, but that this effect is independent of SMC1A phosphorylation and independent of DNA damage. This observation seems confusing relative to their model and requires some explanation.

The reviewer is correct. According to Figure 2a, there is a 1.02-fold change difference between the undamaged WT sample and the ATMi-treated sample(s). While statistically significant, this change is minor compared to the 0.78-fold difference in intra-TAD

contacts between WT and RAD21- samples (also shown in the same panel). We currently lack a clear explanation for this effect. We have discussed these possibilities in the results. In line with this scenario, SMC1A phosphorylation-deficient samples behave similarly to WT (Figure 2a, main text). Furthermore, multiple cohesin subunits are phosphorylated in mammalian cells[11], but their specific effects on cohesin extrusion remain largely unknown.

8. The last sentence in the discussion states that “.....inhibitors of cohesin phosphorylation could be used to sensitize cancer cells to chemotherapeutics similar to ATM inhibition but ideally with reduced toxicity”. It is not clear what the justification for this sentence is, particularly the comment about reduced toxicity. As such, it should probably be removed. Similarly, the previous sentence about A-T and ATR-X is confusing – while A-T is caused by ATM deficiency, ATR-X is a totally different disease; were the authors thinking that it was caused by ATR deficiency (it is not).

We agree with the reviewer that these sentences were confusing. We have therefore removed them from the manuscript.

Reviewer #3

I have read the manuscript “A Dual Role of Cohesin in DNA DSB Repair” By Michael Fedkenheuer et al with initial interest and curiosity. The authors set out to decipher the roles of Cohesin mediated chromatin loop extrusion and the role as a local DNA repair factor via phosphorylation of the Cohesin subunit Smc1 for DNA double strand break (DSB) repair, and claim that indeed two discrete roles for the separate functions of Cohesin can be understood from their results. After having read the manuscript carefully I am not completely convinced, but this could also be due to the possibly not optimal presentation of the data in the manuscript.

A major concern I have is the procedure and controls for degradation of Rad21 through a mAID degenon system. The cells used are created and published and that is all fine. However I strongly recommend to show the degree of degradation in the hands of the investigators. It is claimed in the method section that this is done routinely by FACS, thus there is no reason to not show the results. Preferably for all experiments where this method is used.

We appreciate Reviewer 3's comments on this point. To provide further clarity, we have included Figure S2a, which illustrates RAD21 depletion in both WT and Ligase 4^{-/-} cells. This figure also features an APA analysis that assesses loops before and after RAD21 depletion, confirming the global loss of architecture. Additionally, we have added immunofluorescence microscopy panels for the mES degenon in Figure S2f.

Figure S2a: RAD21 can be efficiently degraded by the addition of indole 3 acetic acid (IAA) for 6h (top panels) which we measured via loss of RAD21-GFP signal. This degradation leads to inactivation of the cohesin complex resulting in loss of genomic loops visualized by APA analyses (bottom panels).

Figure S2f left panel: The left panel shows an example of RAD21 depletion in mES cells using the same concentration and timing as HCT116. Since RAD21 is fused to a Halo tag, these cells were imaged with the Janelia Fluro 549 ligand.

Furthermore, although the advantage of the AID system over the siRNA or RNAi based removal of target proteins has been established lately it is still important to make sure that your cells don't show phenotypes based on the addition of Auxin alone. This could be done by the addition of Auxinole in parallel to Auxin. Or Auxin to cells without the AID tag on the target gene. If this is done or not is not clear from the figures, Figure legends or methods. This should be clarified.

We appreciate the reviewer for highlighting this issue. Auxin has been used in several DNA damage-related studies at similar concentrations without any reported side effects[12, 13]. However, to strengthen that conclusion we have included additional controls in the new version of the manuscript.

First, we aimed to determine whether auxin or RAD21 depletion induces double-strand breaks (DSBs). This control is included in Figure 2c, showing RAD21 depletion at 0 hours in HCT cells, and in Figure S2f for mES cells. We have also added a no DSB control for RAD21 depletion with auxin in Figure 3b, demonstrating that neither treatment causes AsISI-based deletions.

Second, we wanted to ensure that auxin does not affect DSB repair. While having the auxinole control would be ideal, the concentration required to counteract the 750 μ M auxin (approximately 2.5 mM) induce significant toxicity and cell death. To rule out that auxin affects repair independently of cohesin, we have also included a WT control (no RAD21 degron, no OsTIR1) below.

Figure R4: HCT116 cells without components of the degron system were used as a control for delayed repair assays. These cells were treated with 500 μ M auxin for 6h prior to treatment with 2.5 Gy γ -irradiation. There is no difference in repair between these conditions

A second important concern is that the authors use the AsiSI system in some and induction of DNA damage through γ -radiation in other experiments. These two are not all comparable. While AsiSI presumably creates clean DSB γ -radiation certainly does not, rather all types of damage including SSBs and crosslinks. In addition, even damage to proteins can occur. The shift from one system to the other should be indicated in the text and motivated for each experiment.

We appreciate the reviewer for bringing this oversight to our attention. To address this, we have revised the text in the results section to clarify when certain experiments are not feasible using a continuous DNA damage system. Additionally, we have included more details about the COMET assay methods and specified that we utilized neutral conditions to visualize DNA damage specifically resulting from DSBs.

In addition to my major concern there is a number of smaller and bigger issues that I believe should be handled to improve the manuscript. I list them below as they appear in the manuscript with the line/row number as the indicator for each:

7. The address of the first and corresponding author looks strange/repetitive.

Thank you for catching this error. We have deleted the repetitive text.

35-36. Do you want to use numbers in the text in the abstract? Probably better with First and Second. Since I don't think that you have necessarily shown that the two functions are independent you might also consider reformulating the abstract. More on that below.

-We have changed numbers to text and reworded the abstract. The changes should reflect the message we are trying to convey more clearly.

46. Check your references throughout, at least 4 and 6 are not about DSBs or DSB repair.

We thank the reviewer for pointing this out. these citations have been removed.

48-49. formulation; the sentence “cohesin also regulates the topology of chromosomes by extruding through chromatin and forming DNA loops” sounds as if cohesin is extruding through chromatin. I believe it is chromatin that is extruded through cohesin.

We have clarified the wording in the suggested manner.

53. Regarding Refs 13-16, you could refer to the first papers here instead. Also, you could introduce Cohesin associated proteins such as NIPBL and PDS5 here since you are later talking about them.

We thank the reviewer for this suggestion; however we are unable to revise this section in a manner where citations and ideas are clear. We introduce them instead as two separate points for clarity.

64. “arereadily” should be separated and “disperse” should be dispersed, I assume.

Yes, Fixed.

67. You cannot really say “cohesin extrusion” maybe cohesin based loop extrusion or sth similar.

We have made the suggested change.

Worth mentioning already here is that also Smc5/6 is performing loop extrusion and is essential for DSB repair. Could potentially be brought up in the discussion.

We thank the reviewer for this suggestion and have tried to add this point in, however, it overcomplicates the discussion.

71. Segregation between TADs?

We have clarified the wording.

76. The statement MRN activates ATM should be referenced.

Cited

78. Reference 35 is a yeast paper –better to refer to a human study I think.

Re-cited with a human study

87-88. Sentence: "To address this question we here study DNA damage in cells where cohesin is depleted by a degron system and phosphorylation deficient mutants" I would write: study DNA damage and repair ...either by a degron system....or in cells harbouring a phosphorylation mutant. Mutant in singular.

Agreed, Fixed.

91. insert time after same

Fixed.

92. replace both with these. If the functions are distinct but complementary have you then really separated them in their importance for DSB repair?

Fixed and adjusted language a bit

98. spell out ER first time

Fixed.

99-100. Show the IF in the absence of 4OHT (or presence of solvent) treatment as negative control.

The negative control (4OHT-) is shown on the leftmost panel.

101. In Fig S1a there is no location shown for the AsiSI breaks. Should also Fig 1b be mentioned? 102. Also difficult to understand the numbers. It says 80-85 AsiSI breaks in the text in Fig S1a it says 1219 total breaks. I understand that these are the potential sites due to sequence - but this could be better explained. Also for Fig 1 (and all) spell out abbreviation NT once.

We added supplemental Figure S1a to illustrate the consistency of AsiSI breaks among genotypes in three full chromosome examples. It is also apparent that Ligase 4 and SMC1 (to a lesser degree) have increased MRE11 signal indicating that a higher proportion of the cell population have unrepaired breaks at those sites. Also, NT has been changed to no treatment (NT) in Figure 1b which is the first occurrence.

Figure S1a: MRE11 breaks were consistent among genetic backgrounds as shown in three full chromosome examples. MRE11 signal localized primarily to unmethylated AsiSI sites (red).

121. An alternative for investigation of where at the DSB Cohesin loads and ends up, is to utilise an Smc3-ATP hydrolysis mutant. This type of Cohesin mutant was shown to accumulate at DSB ends as opposed to wt cohesin that is found at its normal binding sites (Scherzer et al 2022).

We appreciate the reviewer's suggestion. We attempted to overexpress extrusion-deficient SMC3 but encountered challenges in achieving an acceptable signal-to-noise ratio. To provide further insight, we have included a more detailed view of the break site in Figure S1c. While we do observe some diffuse cohesin signal around the DSB, it does not appear as distinct peaks.

125. I assume you want to say Figure 1f instead of 2b here.

Yes, we have now corrected that. The text references the correct figure now. Note- Due to changes in the figure layout, this figure is now correctly referenced as figure S2b.

137. replace cohesin extrusion with cohesin based loop extrusion or sth similar

Fixed.

138. Introduce the Rad21- cell line, confirm that it is healthy, that degradation of Rad21 works what is the time required for degradation. Is it complete? And what is your control WT cells plus Auxin or Rad21 AID cells plus Auxinol?

Figure S2a was added to validate cohesin degradation. In Figure 2c, the 0h (non-irradiated) time point for RAD21-, WT, SMC1 is shown. This demonstrates that our conditions and/or treatments do not produce DNA damage. Additionally, we added no damage control to Figure 3b.

146. increase in inter chromosomal contacts?

We clarified this in the text. These are intrachromosomal interactions between all DSBs within a chromosome. This was done for each chromosome individually and compiled. The inter-chromosomal analysis is too computationally expensive to be analyzed in a realistic time.

157. Motivate why you abandon the specific DSB system for γ -radiation due to the reasons as mentioned above. Also when you use γ -radiation use a unit for radiation that can be understood by most. What does 2.5 γ mean?

We thank the reviewer for catching this oversight. We have changed all instance of “2.5” γ to “2.5 Gy γ -irradiation”

163- Are you here using AsiSI breaks again. Then you are comparing clean DSB with all types of damages. Thus, you have not really tested if γ H2AX requires loop extrusion – or I need a more clear explanation here.

We have clarified this further by analyzing foci after AsiSI is induced. We find that γ H2AX foci are present when cohesin is depleted.

Figure R5: The left panel shows foci counts after AsiSI activation with 1 μ M 4OHT for 4h. The right panel shows the GFP profile for these same cells. Cells were fixed, stained and analyzed for γ H2AX foci. In each cell, RAD21-GFP, γ H2AX (A555), and DAPI images were collected and used to produce these plots.

173. Is the degron system in ES cells the same as used previously in the paper is the degradation of Rad21 comparable. There is no description of the ES cells in the methods as stated.

Cohesin depletion in mES is now shown in Figure S2f. We have also clarified the conditions used for depletion in mES cells were the same as those used in HCT116 cells.

176. reference 45 – isn't that what you also see and show in fig S1f then maybe indicate that.

Thank you. We referenced the appropriate figure in addition to the reference 45.

187. I am not sure you can call Palbociclib a G1 checkpoint inhibitor. It is a CDK4/6 inhibitor and without CDK4/6 activity the cells will arrest in G1. If you inhibit a checkpoint it rather sounds like the cells would continue through the cell cycle.

There is ongoing debate regarding the exact mechanism; however, the predominant model suggests that CDK4/6 plays a catalytic role in phosphorylating the retinoblastoma protein (Rb) during G1 phase. Inhibition of CDK4/6 prevents Rb phosphorylation, leading to repression of E2F-mediated transcription. As a result, cells are unable to express the genes necessary for cell cycle entry[14, 15].

We do recognize that this point may require further clarification, so we have included a comparison of EdU incorporation between cycling and non-cycling HCT116 cells in Figure R4.

Figure R6: Edu incorporation for cycling versus non-cycling cells.

224. I recommend to be slightly more humble and modify demonstrate.

We revised this sentence accordingly.

230- I am not sure that you have separated the loop extrusion function and the P-repair factor function. I need better and clearer arguments for that. Previous work has before shown that Cohesin protects from repair between distal DSBs (Gelot et al 2016) this should at least be discussed.

This work is now cited and discussed in the introduction. We thank the reviewer for pointing this out.

234- you say that siRNA systems lead to incomplete knockdown, but as long as you have not shown the degree of degradation of Rad21 - your system is not better. Also, I have not seen a test of the actual loop pattern after Rad21 degradation in your study, making the comparison difficult.

The reviewer is correct; that particular sentence in the original manuscript lacked clarity. We have now revised it to indicate that the discrepancy between our results and those of Legube and colleagues may be attributed to several factors, including the relative effectiveness of RAD21 depletion via siRNA compared to the degron system. In our response to Reviewer 3's initial comments, we have also included histogram plots illustrating the extent of cohesin degradation, as well as APA analyses of contacts between TAD boundaries in the presence and absence of cohesin.

You look at intra and inter-chromosomal interactions, but you don't show changes or no changes in loop patterns/positions. In relation to this the paper by Piazza A et al 2021 could be considered to widen the discussion.

We appreciate the reviewer's excellent suggestions. We have added an analysis of gained and lost loops in the results section. Additionally, the paper by Piazza et al. (2021) supports one of our primary hypotheses, which we discuss at the end of the discussion section and have appropriately cited.

289. However cohesinopathy patients don't seem to have an increased risk of Cancer.

That is correct, we have rectified the sentence in question.

Fig 2c. looking t 1h – you might need early events.

As shown in the modified figure (reproduced below), we have found no major difference in foci at early time points.

Figure R7: A short time point experiment was done at 0, 15, 30, 45, and 60 minutes post treatment of HCT116 cells with 2.5 Gy γ -irradiation. The top two panels show foci intensity in WT and RAD21- cells. The bottom two panels show the foci counts. The final plot shows the cumulative GFP intensity summed between time points to confirm RAD21 depletion.

Fig 2d. What is the dosage used for damage in the Comet assay? Is the comet assay alkali based or neutral in pH? This decides what type of damage you can say sth about.

We thank the reviewer for catching this. We forgot to add the methods for the COMET assay. We further adjusted the text in the results and figure legends to specify the conditions clearly. The assay is done under neutral conditions.

Fig 3b. Describe the detection of deletions between breaks better. How big deletions?

What is regarded as a deletion? It would also be nice to see a primer pair used that gives 100 % signal when there is no damage induced for comparison.

We have included Figures S3a-c to demonstrate the specificity of the qPCR assay. The deletion junctions are already ordered by size, and we have also provided the exact deletion sizes in the methods section, as detailed in Table 1.

Fig S2f. Where is the signal for no DSB?

We have included KAP1 phosphorylation data for mES cells at 5h post γ -irradiation, including the proper controls.

For all of my comments it might be me that have not understood. In such case I apologize. However that means that you need to explain better.

We believe that all reviewers' questions and requests have improved the clarity of the manuscript.

References

1. Kouzine, F., et al., *Global regulation of promoter melting in naive lymphocytes*. Cell, 2013. **153**(5): p. 988-99.
2. Arnould, C., et al., *Loop extrusion as a mechanism for formation of DNA damage repair foci*. Nature, 2021. **590**(7847): p. 660-665.
3. Arnould, C., et al., *Chromatin compartmentalization regulates the response to DNA damage*. Nature, 2023. **623**(7985): p. 183-192.
4. Caron, P., et al., *Cohesin protects genes against gammaH2AX Induced by DNA double-strand breaks*. PLoS Genet, 2012. **8**(1): p. e1002460.
5. Gelot, C., et al., *The Cohesin Complex Prevents the End Joining of Distant DNA Double-Strand Ends*. Mol Cell, 2016. **61**(1): p. 15-26.
6. Natsume, T., et al., *Rapid Protein Depletion in Human Cells by Auxin-Inducible Degron Tagging with Short Homology Donors*. Cell Rep, 2016. **15**(1): p. 210-218.
7. Kong, X., et al., *Distinct functions of human cohesin-SA1 and cohesin-SA2 in double-strand break repair*. Mol Cell Biol, 2014. **34**(4): p. 685-98.
8. Kinner, A., et al., *Gamma-H2AX in recognition and signaling of DNA double-strand breaks in the context of chromatin*. Nucleic Acids Res, 2008. **36**(17): p. 5678-94.
9. Sharma, S., et al., *Homology and enzymatic requirements of microhomology-dependent alternative end joining*. Cell Death Dis, 2015. **6**(3): p. e1697.
10. Kitagawa, R., et al., *Phosphorylation of SMC1 is a critical downstream event in the ATM-NBS1-BRCA1 pathway*. Genes Dev, 2004. **18**(12): p. 1423-38.
11. Bass, T.E., et al., *ATM Regulation of the Cohesin Complex Is Required for Repression of DNA Replication and Transcription in the Vicinity of DNA Double-Strand Breaks*. Mol Cancer Res, 2023. **21**(3): p. 261-273.
12. Mirman, Z., et al., *53BP1-shieldin-dependent DSB processing in BRCA1-deficient cells requires CST-Polalpha-primase fill-in synthesis*. Nat Cell Biol, 2022. **24**(1): p. 51-61.
13. Wu, J., et al., *Cohesin maintains replication timing to suppress DNA damage on cancer genes*. Nat Genet, 2023. **55**(8): p. 1347-1358.
14. Pennycook, B.R. and A.R. Barr, *Palbociclib-mediated cell cycle arrest can occur in the absence of the CDK inhibitors p21 and p27*. Open Biol, 2021. **11**(11): p. 210125.
15. Crozier, L., et al., *CDK4/6 inhibitor-mediated cell overgrowth triggers osmotic and replication stress to promote senescence*. Mol Cell, 2023. **83**(22): p. 4062-4077 e5.

I thank the authors for making the extra-effort to address my last concerns. I have to say that I am surprised at the outcome of the average HiC contact matrix analyses (figure 3 of rebuttal), not only the authors are not detecting “stripes” in their HiC datasets (hence there is no evidence of loop extrusion in their experiments) but, at least judging from the figure, it looks to me that they failed to detect stripes also in the re-analysed data from Arnould, et al. (Loop extrusion as a mechanism for formation of DNA damage repair foci. Nature 590, 660–665 (2021). <https://doi.org/10.1038/s41586-021-03193-z>), contrary to what shown in figure 2B of this paper.

We clearly see the stripes described by Arnould et al. (right panel-indicated by white arrows) when HiC data deposited with their manuscript was re-analyzed. This is not a point of contention. The signal strength may look weaker than expected because we only used one replicate to generate this composite. We did not observe these DSB specific stripes in the HiC data we produced. This could be due to differences in resolution, strength of DSBs, or physiological differences between cell lines. Thus, we don't deny that these stripes can occur, but we hesitate to ascribe physiological meaning to this effect.

It is also perplexing that the re-analyses of ChIP-seq data from Arnould et al. revealed minimal accumulation of cohesin at DSBs.

In their manuscript, Arnould et al. also reported minimal recruitment, as evidenced by the values on the y-axes. For instance, in Figure 4b, the enhanced signals for SCC1 and pSMC3 around a DNA break are modest and are most apparent when subtracting the undamaged control signal. This observation aligns with our findings, which indicate that cohesin loading and intra-domain interactions exhibit minimal changes in response to DNA damage.

I have no additional comments.

We thank the reviewer for the hard work they have put in to improve this manuscript.